# Preclinical Models and Technologies in Glioblastoma Research: Evolution, Current State, and Future Avenues

**DOI:** 10.3390/ijms242216316

**Published:** 2023-11-14

**Authors:** Hasan Slika, Ziya Karimov, Paolo Alimonti, Tatiana Abou-Mrad, Emerson De Fazio, Safwan Alomari, Betty Tyler

**Affiliations:** 1Department of Neurosurgery, Johns Hopkins University School of Medicine, Baltimore, MD 21287, USA; hslika1@jh.edu (H.S.); dr.ziya.karimov@gmail.com (Z.K.); salomar1@jhmi.edu (S.A.); 2Faculty of Medicine, Ege University, 35100 Izmir, Turkey; 3School of Medicine, Vita-Salute San Raffaele University, 20132 Milan, Italy; p.alimonti@studenti.unisr.it (P.A.); e.defazio@studenti.unisr.it (E.D.F.); 4Faculty of Medicine, American University of Beirut, Beirut P.O. Box 11-0236, Lebanon; tfa14@mail.aub.edu; 5Department of Neurosurgery, University of Illinois at Chicago, Chicago, IL 60612, USA

**Keywords:** glioblastoma, preclinical models, animal models, genetic engineering, cell lines, microfluidics, 3D models

## Abstract

Glioblastoma is the most common malignant primary central nervous system tumor and one of the most debilitating cancers. The prognosis of patients with glioblastoma remains poor, and the management of this tumor, both in its primary and recurrent forms, remains suboptimal. Despite the tremendous efforts that are being put forward by the research community to discover novel efficacious therapeutic agents and modalities, no major paradigm shifts have been established in the field in the last decade. However, this does not mirror the abundance of relevant findings and discoveries made in preclinical glioblastoma research. Hence, developing and utilizing appropriate preclinical models that faithfully recapitulate the characteristics and behavior of human glioblastoma is of utmost importance. Herein, we offer a holistic picture of the evolution of preclinical models of glioblastoma. We further elaborate on the commonly used in vitro and vivo models, delving into their development, favorable characteristics, shortcomings, and areas of potential improvement, which aids researchers in designing future experiments and utilizing the most suitable models. Additionally, this review explores progress in the fields of humanized and immunotolerant mouse models, genetically engineered animal models, 3D in vitro models, and microfluidics and highlights promising avenues for the future of preclinical glioblastoma research.

## 1. Introduction

Glioblastoma (GBM) is the most frequently encountered malignant primary central nervous system (CNS) tumor, accounting for 50.1% of all malignant primary CNS tumors and 14.2% of all primary CNS tumors [1]. The annual incidence of glioblastoma is around 35 per million individuals, with a male-to-female ratio of 1.6:1 [2,3]. This incidence rate increases with age to reach around 130 per million individuals in the ≥65 years age-group [3]. It is classified as a WHO grade IV tumor under the family named “Adult-type diffuse gliomas.” The current initial management of GBM includes maximal safe surgical resection followed by six weeks of concurrent radiotherapy and temozolomide (TMZ) chemotherapy. After that, patients are maintained on six cycles of TMZ [4,5]. In addition, Gliadel^®^ wafers can be placed in the surgical resection cavity for a sustained local release of carmustine [6,7]. Despite this aggressive multifaceted management, the median survival for patients with GBM is 15 months, and the 5-year survival rate is less than 10%. Nevertheless, several factors affect the specific prognosis of patients and their response to treatment. Notably, age, Karnofsky performance score, and extent of surgical resection are important prognostic factors. In addition, certain molecular markers of the tumor have been shown to play critical roles. For instance, isocitrate dehydrogenase (IDH)-wildtype tumors show a more aggressive phenotype and are associated with lower overall survival rates. In addition, the methylation status of the O6-methylgunaine DNA methyltransferase (*MGMT*) gene has been proven to correlate with response to treatment with TMZ [4]. The poor prognosis of patients with GBM and the lack of definitive treatment regimens have motivated researchers to investigate novel efficacious agents and modalities. However, despite this momentum, no new therapeutic agents have been approved for the treatment of newly diagnosed GBM since the approval of TMZ in 2005 [8]. This has also been reflected by the high failure rates of phase III clinical trials and their inability to recapitulate the efficacy of agents that is observed in previous phases or in preclinical experiments [9,10,11,12].

Hence, representative preclinical models that can faithfully mirror the characteristics of human GBM, its response to treatment, and the myriad of interactions it has with its microenvironment and the immune system are urgently needed and have been the focus of several institutions worldwide. This review aims to describe the evolution of in vitro and in vivo preclinical models over the past decades. Further, it focuses on the most widely used models at the present time and critically details the strengths and shortcomings of each. Moreover, it delves into the emerging fields of 3D in vitro modeling and microfluidics, which have augmented the abilities of in vitro GBM research and constitute promising methods for future cost-effective and time-saving investigations. Overall, this review offers a holistic and comprehensive picture of the available models, appropriate ways of utilizing them for glioblastoma research, and potential avenues for advancing them.

## 2. Methods

We performed a search of the English literature on the topic using PubMed, Ovid Medline, and Google Scholar on 1 April 2023 which was updated on 3 November 2023. The search terms used are “preclinical”, “animal models”, “cell lines”, “in vitro”, “in vivo”, “laboratory animals”, “bench-top”, “investigative techniques”, “three-dimensional”, “bioprinting”, “microfluidics”, “culture techniques”, “animal experimentation”, “glioblastoma”, “glioma”, and “brain cancer”, combined using the articles “AND/OR”, as appropriate. The obtained results were screened for eligibility and relevance using the titles and abstracts. The reference lists of the included articles were also screened for the further inclusion of relevant publications. We used broad search terms and multiple databases to minimize publication bias; however, bias might still be present due to the exclusion of non-English literature, conference proceedings, and unpublished findings.

## 3. Cell Lines Used in GBM Research

Immortalized cell lines are one of the major pillars in GBM research and play major roles in both in vitro and in vivo studies. These lines are typically created by exposing the animals to chemical/oncogenic substances or extracting the cells from human patient tumors and cultivating them in various aqueous and enriched environments. In this context, we will discuss some of the currently used established cell lines and the advantages and disadvantages of each (summarized in Table 1).

### 3.1. C6 Cell Line

The C6 cell line is one of the first immortalized cell lines that was introduced in the 1960s and one of the most preferred in GBM research. It was developed using N-methyl nitrosourea exposed astrocytes of outbred Wistar rats for eight months [13,14,15]. These cells have a wild type of the *Tp53* gene. Schlegel et al. reported that nitrosourea-induced rat glial cells usually lost their *p16/Cdkn2a/Ink4a* gene locus homozygously [16]. The platelet-derived growth factor (PDGF)-β, epidermal growth factor receptor (EGFR), insulin-like growth factor (IGF)-1, Erb3/Her3 precursor proteins, and *Rb* gene have increased expression. Moreover, the TGFα precursor, but not the *TGFα* gene itself, is overexpressed [14,17]. These features are similar to those of human glioma cells [45,46,47]. The reduced expression of fibroblast growth factor 9 (FGF-9) and FGF-10 has also been reported [17]. In addition, there is an increase in Ras activity similar to that observed in human glioma cells [17,48]. Different from human glioma, IGF-2 has decreased expression [17,49]. Another shortcoming of this cell line is that IDH-1 and IDH-2 mutations are not present in the C6 cell line [14,50,51]. A lack of IDH-1 and -2 mutations decreases the C6 cells’ sensitivity to chemotherapy. However, studies reported that an artificial mutation of IDH2 can be introduced, and it rescues the properties of cell migration and makes the cells more sensitive to chemotherapy [18,52]. Unlike human GBM, the C6 cell line is glial fibrillary acidic protein (GFAP)-positive [19,53].

Five to seven days after the implantation of C6 cells, a T2-weighted MRI showed an induced GBM model in the rat brain [54,55]. The main disadvantage of this line is the triggering of the allogeneic immune response [19]. For that reason, the C6 line has limited utility in immunotherapy studies in Wistar and BDX rats [15,21]. Parsa et al. reported that CD3+ lymphocytes were detected in the C6/Wistar model in flank tumor and brain tumor samples on the 10th day and 12th day, respectively [21]. The study also found that this model had significantly high anti-glioma antibodies, especially higher in the flank group. Studies reported that the tumor had well-demarcated borders resembling metastatic lesions rather than primary glioma when it was used in the Sprague-Dawley and Long-Evans rats [56,57]. C6-derived tumors rely on the secretion of vascular endothelial growth factor (VEGF) and basic fibroblast growth factor (bFGF) to drive neovascularization [22,58]. The C6 cell line has been widely used to assess the effectiveness of antiangiogenic therapy, chemotherapy, radiotherapy, oncolytic viral therapy, gene therapy, photodynamic therapy, and treatment with proteosome inhibitors [59,60,61,62,63,64,65,66,67].

### 3.2. 9L Cell Line

The 9L cell line was developed by the intravenous administration of 5 mg/kg of N-methyl nitrosourea to Fischer rats for 26 weeks [19,23,24]. The 9L cells have sarcomatous histology when they are implanted intracerebrally in rats [17]. For that reason, 9L-derived tumor models were also called gliosarcoma. TGFα and EGFR are overexpressed. There is a reduction in the expression of FGF-2, FGF-9, FGFR-1, and PDGF-β [17]. This tumor model has high immunogenicity that reduces its utility in immunotherapy studies [15,19]. The model possessed a mutant *Tp53* gene [68]. This cell line has broad usage in evaluating PET and MRI studies that investigate tumor hypoxia and metabolism, the mechanism of chemotherapeutic resistance, drug transportation through the blood–brain barrier (BBB) and blood–tumor barrier, and the effectiveness of antiangiogenic treatments, chemotherapy, radiation therapy, boron neutron capture therapy (BNCT), immunotoxin therapy, gene therapy, and oncolytic viral therapy [69,70,71,72,73,74,75,76,77,78,79,80,81,82,83,84,85]. An examination of rat tumor models with 9L cells showed infiltration with CD8+ T lymphocytes as proof of an antitumoral immune response [15].

### 3.3. F98 Cell Line

The F98 cell line was developed by the intravenous administration of 50 mg/kg N-ethylnitrosourea to pregnant Fischer rats on the 20th day of gestation [15,19]. It has an anaplastic appearance, which resembles GBM, and a spindle cell histological pattern [19]. This line has an increased expression of PDGF-β, Rb, Ras, EGFR, cyclin D1, and cyclin D2 and positive staining for Vimentin and GFAP [17,25]. The F98-derived tumor has weak immunogenicity with insignificant CD3+ T cell infiltrate, and this characteristic makes it useful for immunotherapy studies [25]. This cell line is refractory to carboplatin and paclitaxel therapy and has a poor response to photon radiation [19,86,87]. Also, decreased breast cancer 1 (*BRCA1*) gene expression was reported in this line, which causes a disrupted DNA repair and decreased response to some chemotherapeutics and photon therapy [71]. However, F98 cells showed a good response to BNCT and to 6 Megavolt (MV) radiotherapy with the combination of carboplatin or cisplatin administered intracranially by convection-enhanced delivery [86,88]. Studies reported that these cells possess a high invasive potential and were found at distant loci in the primary tumor and perivascular area [15,19]. Moreover, a high mitotic index, a necrotic core, and neovascular proliferation were reported [25]. F98 cell-derived glioma is used to assess the efficacy of radio-iodine therapy, iodine-enhanced synchrotron stereotactic radiotherapy, tumor angiogenesis, the molecular targeting of EGFR, diffusion tensor imaging, chemotherapeutics, such as tonabersat, Nitrone OKN-007, and trimetazidine, suicide gene therapy with Herpes Simplex Virus 1—Thymidine Kinase (HSV-TK), and the liposomal formulation of carboplatin [89,90,91,92,93,94,95,96,97].

### 3.4. RG2 Cell Line

This cell line was generated through the intravenous administration of 50 mg/kg N-ethylnitrosourea to pregnant Fischer rats on gestational day 20 [19]. Due to the similar generation process, it shares several similarities with the F98 cell line. RG2 cells exhibit a very invasive pattern, which makes it a suitable model for GBM [98]. Sibenaller et al. reported that this line is refractory to several therapeutic methods, which also makes it a good model for GBM [17]. The RG2 cell line has increased expression of PDGF-β, IGF-1, Erb3/Her3 precursor proteins, Ras, and D2. They possess the wild-type *Tp53* gene and a loss of expression of the *p16/Cdkn2a/Ink4* gene locus [16,17]. This line is used to evaluate chemotherapy, radionuclide therapy, antiangiogenic therapy, gene therapy, vascular permeability, BBB permeability, and oncolytic viral treatment [99,100,101,102,103,104,105,106,107]. RG2 cells are very weakly immunogenic in syngeneic Fischer rats and have attenuated expression of the major histocompatibility complex (MHC) class I antigen [98]. However, in vitro IFN-γ treatment of the cells causes increased MHC-I expression, and the in vivo high-dose (2.4 × 10^5^ U/kg) IFN-γ intracarotid infusion treatment of Fischer rats showed a remarkable qualitative antitumoral immune response against RG2 cells and increased survival at 34 days [108]. This cell line was also used to confirm the sonosensitizing effects of 5-aminolevulinic acid in vitro [109]. Notably, despite their abundance, rat cell lines are not commonly used in immunotherapeutic studies due to the paucity of monoclonal antibodies designed to target rat antigens and immune surface markers [15]. Moreover, there are fewer genetically engineered rat glioma cell lines as compared to mouse ones [15].

### 3.5. GL261 Cell Line

The GL261 cell line was developed by the intracranial injection of 3-methylcholantrene into C57BL/6 mice [26,110]. This line possesses mutations in p14, p16, phosphatase and tensin homolog (PTEN), K-ras, and EGFR [111]. It is negative for GFAP but positive for Vimentin [27]. In addition, decreased H-ras expression has been reported [26]. Moreover, IDH1 is wild-type [29]. GL261 cells have detectable levels of expression of MHC class I, which makes them moderately sensitive to Natural Killer cells [26,30,31]. On the other hand, the expression of MHC class II was not detected but can be increased by stimulation with IFN-γ [26,31]. The GL261 cell line was described as a weak immunogenic line, which makes it a good option for immunotherapy studies, such as anti-PD1, anti-CTLA4, and dendritic cell therapy, gene therapy, and cytokine-secreting vaccine studies [30,32,112,113,114,115,116]. The tumor model, both in its subcutaneous and intracranial forms, usually does not metastasize; however, it has a rapid growth pattern [26]. Maes et al. reported that one of the limitations of these cells is that they have distinct growth motifs, which differ from the irregular growth of the high-grade glioma that restricts tumor resection in animal models [31]. On a histological level, GL261-induced tumor models showed pseudo-palisading necrosis, perivascular proliferation, and nuclear pleomorphism [32,33]. The GL261 tumors also show histological similarity to ependymoblastomas [29]. In vitro, GL261 cells were sensitive to low-dose (2 Gy) radiation therapy; however, in vivo, radiotherapy with a 4 Gy dose decreased the tumor growth of implanted cells rate but did not prolong survival [26]. In addition, these cells exhibited resistance to TMZ in an in vitro study [117]. The GL261 cell lines are almost exclusively used in immune studies. For instance, the immune checkpoint inhibitor anti-programmed death-1 (anti-PD1) was shown to have a synergistic effect with BHV-4157, a glutamate regulator, using a glioblastoma mouse model with implanted GL261 cells [118]. 

### 3.6. CT2A Cell Line

The CT-2A cell line was developed in C57BL/6 mice by induction with methylcholanthrene [29,119]. This cell model showed a high mitotic index, an increased cell density, nuclear polymorphisms, hemorrhagic areas, pseudopalisading necrosis (central necrotic area), high angiogenesis, and microvascular invasion [36]. Invasiveness to the circumambient parenchyma is weak with this cell line showing sharp tumor borders when grown in vivo [34], which is different from the human GBM. Nevertheless, this cell line shows higher invasiveness and migration when compared to other glioma cell lines in vitro [35]. This cell line has wild-type p53 and IDH1 and lacks PTEN expression [35,120,121]. It stains positively for GFAP, which was commonly observed in the tumor edge area and also in the vicinity of small vessel walls within the tumor [36]. In a similar fashion, *SOX9* and *SOX10* genes were expressed in the tumor mass, wherein Sox9-positive tumor cells were localized to the tumor rim and perivascular areas, while Sox10-positive tumor cells were limited to the tumor rim and rarely present in the inner area of the tumor. Additionally, in contrast to the normal brain astrocytes GFAP and Sox9 concomitantly, CT2A tumors show distinct expression patterns for GFAP and Sox9, with some cells expressing both and others expressing only one of the two markers. This suggests that CT-2A tumors have high intratumoral heterogeneity, which poses an additional obstacle in treating these tumors. One of the important pros of the CT-2A cell line is its high tumorigenic burden, with the shortest median survival for mice bearing these tumors compared to other GBM models, which shortens the duration needed to complete in vivo studies [122,123]. This cell line has been utilized for the investigation of different drugs and has been especially useful in immunotherapy studies [15]. In this context, the stimulator of interferon genes (STING) agonist, ADU-S100, was shown to promote the innate immune response against implanted CT2A glioma and subsequently prolong the survival of the hosting mice [124]. Moreover, Barnard et al. generated an oncolytic herpes simplex virus-1 (oHSV-1) vector that expresses fms-like tyrosine kinase 3 ligand (Flt3L) and investigated its efficacy via intratumoral injection into CT-2A glioma-bearing C57BL/6 mice, and this resulted in enhanced survival [125]. 

### 3.7. U87 Cell Line

The human-derived U87 cell line was developed in 1968 [126]. This line harbors mutations in the *p14*, *p16*, and *PTEN* genes [43,44,127]. However, Tp53 is wild-type [128]. The U87 cell line does not express S100 and GFAP, although it derived from a glial origin [42]. On the other hand, Vimentin is positive [41]. It presents with weak vascularization [111]. Low necrotic foci were observed in the intracranial tumor model, unlike the subcutaneous model [40]. In contrast to the human GBM, the U87 cell line presents sharply demarcated tumor borders, which is the main limitation of these cells [110]. Mesti et al. reported that U87 cells display no expression of VEGFR-1, low expression of VEGFR-2, and increased levels of VEGF [39]. IDH1 is wild type [29]. The study showed no expression of CD31, VE-cadherin, Tie1, or Tie2 as endothelial cell-specific markers [129]. However, the cells stained positive for the CD133 marker, which allows for the formation of neurospheres [29,130]. High colony formation and migration potential were shown in vitro [38,43]. A study found moderate Synaptophysin, kallikrein, and CD68 expression in U87 cells [41]. It was hypothesized that this may be related to the mesenchymal cell characterization of the model. Ras pathway activation was observed in this line [131], whereas no EGFR amplification was found [132]. Researchers reported that U87 tumors exhibit an anaplastic pattern [41]. Interestingly, these cells are responsive to TMZ treatment [29]. U87 has been used to evaluate the chemotherapeutic agents, tetrandrine citrate, chloroquine, sirolimus, temozolomide, and bevacizumab [133,134,135].

### 3.8. U251 Cell Line

The U251 cell line is also derived from human GBM and is one of the most commonly used patient-derived cell lines [42]. This line harbors mutations in the *p14*, *p16*, *PTEN*, and *EGFR* genes; it also shows PI3K upregulation and the non-functional mutant *Tp53* gene in U251 tumors [37,44,111]. The cell line is positive for S100, GFAP, and Vimentin [94,111]. It displays fast growth patterns [38]. U251-derived tumor models have a high (>50%) Ki-67 staining ratio and perinecrotic areas that stain positive for hypoxia-inducible factor-1-α (HIF1-α) and Caspase-3 [111]. Tumors are also positive for CD133 [29]. U251 cell-derived orthotopic xenografts had a stunning phenotypic similarity to human GBM, with a diffuse infiltration pattern in the parenchyma and significant necrotic foci, a winding pattern of vascular proliferation, nuclear pleomorphism, hemorrhage, and edema [110,111]. These characteristics resulted in an increase in the usage of the U251 cell line in the last decade. However, differing from human GBM, this model does not show an invasive pattern through the white matter tracts [29]. Moreover, it is responsive to both radiotherapy and TMZ [136,137]. This cell line is usually used to investigate the response to chemotherapeutic agents, such as lomustine, TMZ, and carmustine, and the antiangiogenic agent bevacizumab [29].

Human-derived cell models have shown great utility in investigating selective tumor markers, tumor-specific signaling pathways, apoptosis signaling, angiogenesis, tumor phenotypic behaviors, and prognosis-related factors [41,130]. The main disadvantage of both human-derived cell lines is that they trigger a direct immune response in animal models and require the use of immunosuppressed recipients.

## 4. Three-Dimensional In Vitro Models

Three-dimensional (3D) models of GBM arise from the need to overcome specific and consistent limitations of 2D cultures in modelling both the brain tumor biology and therapeutic response. Conventional culture methods are simple, cheap, fast, and rarely associated with ethical concerns [138]. Nonetheless, they poorly reproduce GBM cell interactions with microglia, surrounding neurons, and other tumor microenvironment (TME) entities that govern the core biological processes of cancer cells such as metabolism, stemness, and invasiveness [138]. Moreover, GBM cells tend to become quiescent under static mechanical forces of plastic 2D cultures [139], accumulate mutations over time, and diverge from their original outlook in vivo [140]. In a similar fashion, patient-derived xenograft (PDX) models also exhibit several limitations pertaining to the immune status of the host and the time-consuming process to generate them. In this regard, 3D GBM models have a superior ability to structurally and biologically reproduce GBM features compared to 2D cultures and PDX models and hold great promise for tumor modeling and therapeutic testing in vitro (Illustrated in Figure 1). 

### 4.1. GBM Spherical Cancer Models

Spherical cancer models (SCMs) are constituted of sphere-like components that are mostly or entirely made of cancer cells [141]. Throughout the literature, they have been often confused with cellular aggregates, but aggregates lack the compaction, geometry, and retention of original cell–cell interactions that SCMs display [142]. Weiswald et al. [142] described four main types of SCMs used in GBM research. The first model is tumorspheres (also called neurospheres). Earlier efforts of neurosphere generation date back to the early 1990s [143,144], and in the early 2000s, several studies demonstrated the successful generation of neurospheres from putative glioma stem cells (GSCs) cultured with neurotrophic growth factors like epidermal growth factor (EGF) and FGF2 [145,146,147,148]. First described by Singh et al. in 2003, they are to date the most common SCMs used in glioma research [149,150]. Tumorspheres are generated by the proliferation of single-cell suspensions, tissue-derived cancer cells, circulating cancer cells, or established cell lines that grow by clonal expansion [149] and in the absence of any non-tumor cells [150]. Galli et al. highlighted how, in the resulting spheroids, GBM cells retained a degree of spatial organization and tissue polarity and were therefore considered the first 3D GBM model [145]. In fact, spheroids can retain rudimental features of 3D tumor tissue, including tumor cell–cell interactions, oxygen and nutrient gradients, and the peculiar GBM histoarchitecture with a necrotic non-proliferating core and an outer proliferating layer [150]. Importantly, these models allow for the retention of cancer stem cells, and single neurosphere-derived cells can generate infiltrating gliomas [150]. They can be grown in either specific gels, media, or sera, but the growth medium appears to play a crucial role in maintaining the neurospheres’ biological features. Gel-embedded systems such as Matrigel or agarose hydrogels can mimic the structural and mechanical properties of the extracellular matrix (ECM) or TME, enabling high-throughput drug-screening and personalized therapeutic testing [150,151]. When not grown in gel, patient-derived neurospheres are alternatively maintained in an enriched EGF/bFGF medium and preferentially in non-laminin-coated plates, to maximize stem cell maintenance [150]. Furthermore, a low seeding density is recommended to prevent neurosphere aggregation [150,152,153]. An interesting phenomenon has been observed when the EGF/bFGF trophic factors are substituted with serum: tumorspheres lose their 3D shape and turn into 2D cultures, whose cells produce only demarcated tumors without infiltrative features. Moreover, GSCs change their appearance in a culture and undergo a reduction in both telomerase activity and stem cell expression markers [152,153]. These findings highlight the technical limitations of neurosphere cultures and therefore require careful consideration when interpreting results obtained with these models. 

The second model is represented by multicellular tumor spheroids (MCTS). Being first reported by Sutherland et al. [154] in 1971 and first implemented for glioma research in 1989 by Mashiyama and colleagues [155], it is the oldest SCM currently in use. In MCTS, a large number of cancer cells, either patient-derived or from commercial cell lines, are scattered in non-adhering cultures and allowed to grow and compact spontaneously [150,156,157,158]. The resulting spheroids measure around 400–1000 μm and display intermediate cell–cell junctions as well as inner-to-outer gradients [156,159]. They do not require specific trophic factors in addition to standard serum-supplemented media, making them easy to obtain [150]. Alternative culture methods include liquid overlay and spinner cultures [160]. The third model features organotypic multicellular spheroids, also called organotypic spheroids (OS). They are obtained directly from non-dissociated ex vivo samples that are then grown in culture until they develop a spherical shape [150]. As a result, they retain the original TME and matrix composition [150], a highly desirable feature in glioma modeling for both brain cancer biology experiments and for several types of therapeutic testing. Lastly, the fourth SCM is tissue-derived tumorspheres, but these have not yet been used in glioma research [142,150]. 

Overall, spheroids accurately model the tissue architecture and cellular interactions of the original tumors from which they are derived. Unfortunately, they fail to reproduce the intratumoral heterogeneity of the parental tumor [161], particularly after long periods in culture. Moreover, being composed entirely of cancer cells, the TME cannot be effectively modeled. Some efforts have been made by co-culturing spheroids with stromal cells [161], but the original cellular composition, vascular architecture, and intercellular dynamics of the TME were still lacking. 

### 4.2. GBM–Brain Organotypic Models (GBOMs)

GBM–Brain Organotypic Models (GBOMs) are 3D human stem-cell-derived models introducing a GBM tumor via either gene editing or co-culturing with GBM cells [138]. The first GBOM was generated by Preynat-Seauve et al., who introduced GBM cells in culture with organoids generated from human embryonic stem cells (ESCs) [162]. Organoids are, by definition, included in the group of organotypic cultures [138] and are defined as self-organizing entities that originate from individual stem cells that grow in vitro to give rise to 3D microscopic structures [150]. In recent years, the development of solid protocols for the generation of brain organoids has revolutionized the study of neurobiology and disease. Several protocols for brain tumor organoids have been described and are currently in use, mainly for tumor invasion modeling and therapeutic screening. Pamies et al. [138] described three different categories of organotypic cultures. The first includes cerebral organoids (also called neoplastic cerebral organoids—neoCORs—by other authors) [163], which are 3D brain models derived from induced pluripotent stem cells (iPSCs) that undergo subsequent oncogenic knock-in to reproduce GBM growth [164]. 

The first brain organoids were produced by Lancaster et al. in 2013. Their model successfully recapitulated normal brain development and differentiation and featured structures similar to cerebral cortices, telencephalon, choroid plexus, and other human brain portions throughout a 1- to 2-month culture process. Subsequent efforts by other authors modified this protocol to integrate tumor cells during the organoid development to mirror GBM pathogenesis. Bian et al. [165] combined a transposase and CRISPR-Cas9 system to induce oncogene knock-in in a 3D in vitro neoCOR model, leading to GBM onset. The GFP tagging of neoplastic cells enabled close monitoring throughout tumor growth and transformation. With a similar strategy, Ogawa et al. [166] induced GBM formation in a cerebral organoid model derived from human ESCs and modified through a CRISPR-Cas9-mediated HRasG12V-TP53 homologous recombination. The resulting model displayed a mesenchymal-like GBM tumor growth at single-cell analysis, and the resulting tumor cells could be transplanted and regenerate invasive tumor masses in fresh brain organoid cultures [166]. 

The second type of organotypic cultures, neural organoids (NOs), represents a slight variation to the previously described cerebral organoid model. NOs are also derived from human ESCs, but they were co-cultured with patient-derived GBM cells, which showed organoid invasion after 1–2 weeks [167]. This model has shown a faithful reproduction of the structural organization and distribution of neural cell markers in the tissue [162]. The co-culturing method was also exploited by Linkous et al., who developed Glioma Cerebral Organoids (GliCO) by culturing patient-derived GFP-tagged GSCs with 3D brain organoids [168]. Different GSC lines displayed unique growth and invasion patterns within brain organoids, including nodular, diffuse, and “honeycomb-like” patterns, likely reflecting intrinsic biological features of the parental tumors [163]. Moreover, this GliCO model enabled the maintenance and recapitulation of the mutation profile of the original tumor, contrary to 2D cultures that tend to lose key molecular aberrations over time [168]. Similarly, Fedorova et al. [169] co-cultured brain organoids with GBM spheroids. The resulting GliCO displayed increased cell migration in the brain organoid due to the addition of ECM proteins in the culture. Further, the model showed maturation and plasticity between different GBM subtypes, pointing to the influence of the healthy brain microenvironment, reproduced in the organoid, on the behavior of GBM cells in the spheroid. This model was accurate and significantly faster to develop compared to most other methods, encouraging further developments. 

Finally, Plummer et al. [170] developed the BrainSphere model, the last category of organotypic culture. In this model, the incorporation of GBM cells occurred by aggregation in co-culture [138], and the resulting spheres were smaller compared to other organoids, around 300–350 μm in diameter. Although lacking the complex structural organization displayed by other organoids, particularly those featuring specific brain structures, this model was deemed reproducible, easily adaptable to high-throughput technologies, and did not generate artifacts during drug testing due to its lack of central necrosis [138].

In another attempt to generate GBM organoids, Hubert et al. [171] embedded small tumor fragments obtained from surgical specimens into Matrigel droplets. The resulting organoids retained the 3D spatial distribution and cellular heterogeneity of the parental tumor, allowed the growth of stem, progenitor, and differentiated cells in the same conditions, and mirrored tumor vulnerabilities to targeted therapy and radiation displayed in vivo [163]. The gradients of GSC density in relation to oxygen levels were conserved and similar to those in tumor tissue, with increased Sox2+ cells in the periphery and, to a lesser extent, in the hypoxic core. Different Sox2+ cell properties were observed according to different locations [171], suggesting that this model enabled the growth of a diverse set of GSCs and could be exploited to study the cellular hierarchies of GBM in vitro [163].

A solid protocol for glioblastoma organoid (GBO) generation comes from the work of Jacob et al. [172], who developed a culture method for generating patient-derived organoids from fresh GBM tissue. These organoids were able to grow and expand in vitro, even after biobanking and thawing. They showed hypoxic and nutrient gradients consistent with the metabolic features of GBM tissue and faithfully mirrored the intratumoral heterogeneity of parental tumors, in terms of the cellular composition, mutation profile, and gene expression. They preserved the original cell–cell interactions within the tumor tissue and predicted the therapeutic response of the parental tumor in vitro with excellent accuracy. Finally, the engraftment of tumor cells from these organoids in mouse models successfully generated invasive tumors. In a later study in the same year, Jacob et al. [173] applied their patient-derived organoid model to in vitro chimeric antigen receptor (CAR) T cell testing. They showed that their organoids retained the endogenous expression of the mutant protein EGFRvIII, overcoming a typical pitfall of long-term 2D tumor cultures. Anti-EGFRvIII CAR T cells, when co-cultured with these organoids, were able to infiltrate them and proliferate, resulting in tumor cell killing. Overall, this model recapitulated the original intratumor heterogeneity and showed promising results in modeling the immunotherapy response. However, it showed a variable success rate according to the IDH profile and the primary or recurrent status of the tumor of origin, a finding that deserves further investigation. Moreover, it demonstrated poor TME retention over time due to the limited lifespan and expansion potential of resident immune cells, as well as the low vasculature and immune-related gene expression in the TME [174]. 

Finally, LeBlanc et al. [175] performed a single-cell genomic analysis of GBM samples and matched patient-derived organoid explants. They showed that organoids, but not gliomaspheres, are able to retain both inter- and intra-tumoral heterogeneity typical of their parental tumors, along with their original distribution of cell states. This finding supports the role of organoids as a reliable tool in reproducing the biology of original lesions, a finding that will be crucial for the future pipeline of therapeutic testing in GBM. For instance, Pedrosa and colleagues, who utilized a GBM organoid to test for the efficacy of photodynamic therapy with 5-aminolevulenic acid (5-ALA), showed that this model simulated 5-ALA uptake, protoporphyrin IX emission, and consequent GBM cell apoptosis in a dose-dependent manner [176]. In another study, Morelli et al. utilized GBOs for metabolic imaging of the tumor’s response to TMZ [177]. This protocol was concordant with the molecular profiling of parental tumors and helped identify a new molecular signature associated with better survival. Mitchell and colleagues exploited a patient-derived GBM organoid model to identify the *WDR5* gene as a targetable epigenetic regulator of cancer stem cell maintenance [178]. Finally, Yang et al. tested the chemical compound safranal in both GBM–brain organoid cocultures and 3D spheroid tumor assays, showing that this agent is effective in inducing GBM cell apoptosis and cell cycle arrest and displays synergetic effects with TMZ [179].

Overall, GBOMs are a versatile tool for GBM modeling and are amenable to a wide range of experimental setups. Lineage tracing experiments with either EdU pulse-chase setups, GFP tagging, or lentiviruses/retroviruses have been successfully deployed to track progenitor cells and their lineage in GBOs [180]. Additionally, genetic modifications such as overexpression, shRNA/siRNA-mediated knockdowns, and Crispr-Cas9-mediated knockout have been implemented in GBOs to study GBM pathogenesis [173]. GBOs represent a safe and effective platform for GBM studies, particularly for studying tumor–TME interactions, invasion, and progression. The 3D features of these models enable appropriate cellular differentiation and GSC maintenance, and their free-floating nature makes them easily transferrable between cultures [138]. 

Current limitations include significant variability among single organoids and organoid lines, poor modeling of the vascular and immune compartments of the TME, as well as potential but difficult adaptation to high-throughput screening technologies [138]. Future studies should focus on improving the representation of the most delicate but important components of the GBM TME, including neoangiogenic blood vessels and immune cells. In their review, Klein et al. [181] have considered the reasons behind the technical challenges of producing immunocompetent GBOMs. First, the limited availability and viability of fresh GBM tissue hinders the ability to obtain organoids with functional tumor and TME cells. Second, the mismatch between the timing of lymphocytes/microphage harvesting (immediate) and organoid growth (1–2 weeks) complicates the maintenance and subsequent addition of these immune populations to GBM organoids. T cells may even be harder to harvest, giving the now-ascertained systemic tumor-induced lymphopenia in GBM patients [182,183]. Appropriate culture conditions need to be established, as well as dissociation protocols, given their influence on tumor–immune cell interactions and immune cell infiltration [184]. The mere addition of murine ECM proteins to the human GBOM may also trigger a non-self-immune activation against murine antigens, potentially hampering the development of the model. Last, the intrinsically low immunogenicity of GBM cells is a limiting factor in modeling their antitumor response without immune cell enhancement.

For these reasons, the trend in research to answer these needs is leaning towards 3D bio-printed cultures, and multiple studies have already shown promising results in modeling both the immune microenvironment and tumor vasculature [161,173]. 

Special consideration should be given in future studies to the interaction between GBM cells and neurons, given the recent appreciation of how tumor cells synapse with healthy neurons and use synaptic proteins such as soluble Neuroligin-3 (sNLGN3) to boost their own proliferation [185]. 

GBOMs may also be used to study spontaneous intracellular ionic spikes that occur rhythmically in GBM tissue and drive tumor proliferation through MAPK and NF-kB pathways, a new finding that can shed light on some underappreciated collateral mechanisms of glioma growth [186].

Modeling the BBB and the blood–tumor barrier will also play a crucial role in establishing the dynamics and efficacy of anticancer therapeutic delivery. In this regard, 3D bioprinted models may be more suitable for building and integrating blood vessels and the complex system of tight junctions and transporters of the BBB than simpler GBOMs [161]. Also, Wang and colleagues have postulated the feasibility of GBOs in testing for neoantigen discovery. This comes from considering previous evidence of the maintenance of the stromal compartment of GBM by GBOs, coupled with the fact that this compartment seems to be the one presenting most of the neoantigens in IDH-mutant glioma models [187]. Finally, the discovery of glioma microtubes as a means of tumor cell–cell interaction and mitochondrial exchange [188] for proliferation deserves further characterization, and GBOs have already offered a suitable platform for dedicated studies on this matter [189]. 

### 4.3. Scaffolds

Scaffolds introduce a 3D structure that helps support cell cultures [150]. Different properties like stiffness, porosity, interconnectivity, and structural integrity also help modulate how cells behave in a model [190]. The characteristics of the models attempt to reproduce the specific features of the ECM of tissues of interest [184]. Scaffolds employ different combinations of constituents of tumor ECM, including hyaluronic acid (HA) and synthetic components [191,192,193].

When comparing 2D and 3D cultures, Lv et al. [194] found that 3D collagen scaffolds created more dedifferentiated tissues with a higher similarity to GBM in terms of culture morphology, had higher levels of resistance to alkylating agents, and had an increased regulation of MGMT. More recent techniques in scaffold fabrication include solid free-form technologies; among them, 3D bioprinting allows for better and more accurate recapitulation of the TME by including both tumoral and non-tumoral cells, as well as the macromolecular components of the ECM [150].

#### 4.3.1. 3D Bioprinting

3D bioprinting is a solid free-form scaffolding technique that employs different bio-inks placed layer by layer following a computer-designed pattern [195,196]. There are two bio-ink types: scaffold-base bio-ink and scaffold-free bio-ink. The scaffold-base bio-ink creates layers of living cells onto soft biomaterials, while the scaffold-free bio-ink aggregates cells without an exogenous biomaterial structure [197]; these cells are then fused and grow in tissues of higher complexity [195]. Although scaffold-free approaches have their advantages, they also have limitations. These techniques can be challenging to replicate, and managing spatial distribution and cell density can pose difficulties. Additionally, these methods may not be suitable for high-throughput applications [198]. 3D model biofabrication techniques include inkjet-based techniques, microextrusion, and laser-assisted bioprinting. In 3D bioprinting, many types of biomaterials with varying viscosities and cell densities can be utilized. This allows for models that more closely resemble the TME by controlling the arrangement of different cell types in a 3D structure [161]. 

Hermida et al. developed 3D models made of a combination of ECM proteins, alginate, stromal cells, and U87-MG GMB cells by the microextrusion technique. These models more faithfully modeled the GBM therapeutic response compared to 2D cultures [199]. Using a similar technique, Dai et al. generated a GMB model with 87% survival of GSCs and elevated proliferation rates immediately after bioprinting. The model also demonstrated the differentiation of glioma cells into diverse neuronal populations and the development of blood vessels in its components. When examining the response to chemotherapy in glioma cell cultures, the 3D model better reproduced GMB’s chemoresistance than the 2D model [197]. This finding suggests that the 3D model may better represent the tumor’s behavior, leading to improved drug testing and treatment development. In addition, according to Wang et al., 3D models seem to better recapitulate the histological and molecular features of GBM, including angiogenesis-related gene expression, the capability for in vitro vascularization, and stemness [200].

Since GBM cells present several distinct molecular subtypes with different behaviors and morphologies [201], preserving their intrinsic characteristics in the GMB models is essential. This process would allow for a more accurate representation of drug responses during screening assays. Heinrich et al. [202] utilized 3D bioprinting to develop a miniature brain composed of GBM cells and macrophages. This approach enabled them to investigate the intricate interactions between GSCs and non-tumor cells. This mini-brain model reproduced the interaction between GSCs and the microenvironment, evidenced by tumor-associated macrophages repolarization and epithelial–mesenchymal transition in GBM. Moreover, the mini-brains significantly increased glioma cells’ growth and invasiveness. The group also evidenced that therapies targeting intercellular communication can decrease the growth of tumors in the mini-brain models. 

In their 2021 study, Neufeld et al. [161] devised a fibrin-based 3D bio-printed model containing GBM cells, astrocytes, microglia, and perfusable blood vessels made of a vascular bio-ink coated with pericytes and endothelial cells. This model faithfully reproduced the interaction of GBM cells with the microenvironment in vitro. Their model was also used in the assessment of traditional chemotherapy drugs as well as biological drugs, immunotherapy, and drugs targeting cell adhesions. Indeed, the 3D model was proven to be superior to the 2D model in capturing the complexity of the TME. Furthermore, the cells that grew in the 3D scaffolds retained the upregulation of the same molecules upregulated in vivo, including oncogenes and typical prognostic biomarkers. In conclusion, the genetic outlook, growth, invasion, and TMZ response in these 3D fibrin-bio-ink-based printed models mirrored those of orthotopic mouse models of GBM. It proved superior to the results of similar assays on 2D cultures, showing great promise for future testing. 

The main aim of 3D printing for GBM has been deepening our comprehension of glioma biology, tumor angiogenesis, invasion, malignant transformation, treatment susceptibility, and screening. These 3D models hold vast potential for GBM research, empowering researchers to precisely manipulate and select specific study variables that align with their research goals. By including more than one cell type in the gels, it is possible to understand cell–cell interactions better, thus allowing scientists to explore the complexity of the communication networks between different cell lines and types. The manipulation of gel properties presents an opportunity to explore the impact of the ECM’s physical characteristics on GBM biology. For example, it could be possible to modify the composition of the bio-ink to obtain a recapitulation of different tissue types [161]. This approach also allows for studying mechanotransduction and other related processes, providing a more comprehensive understanding of the intricate interactions between cells and their environment [150]. 

#### 4.3.2. 4D Bioprinting

Four-dimensional (4D) bioprinting is a novel approach that differs from traditional 3D bioprinting by utilizing stimuli-responsive biomaterials that can dynamically change over time, representing the fourth dimension. This approach seeks to recapitulate the natural physiological processes of living organisms and environments [203,204,205]. Its potential applications include medication delivery, drug screening, and the creation of vascularization models. Notably, it can potentially enhance our understanding of GBM development and treatment, thus offering significant value to the medical community [206,207].

## 5. Microfluidics

A microfluidic cell culture provides an accurate way to manage fluid flow at the microliter and nanoliter levels in specific geometries. This allows for manipulating and examining individual cells, tissue cultivation on automated chips, and expanding to larger cell populations [208]. Modern microfluidic chips often utilize miniature micromechanical membrane valves of poly-dimethylsiloxane (PDMS) to control fluids at the microliter scale with greater efficiency. These valves help control the flow and delivery of media, drugs, and signaling factors to live cells in time and space. Fully integrated and compact devices can be easily created using optical lithography, the meticulous organization of flow and control layers, and parallelized fabrication. This allows for the quick assembly of different channel, chamber, and valve types [208].

With microfluidics, it is possible to control the conditions of cell cultures and manage the timing of fluid flow. This process is possible due to the small geometrical dimensions of the culture, which allow for a laminar fluid flow in the nanoliter and picoliter range [209]. The valves also allow for the precise timing of fluid flow, granting control over the chemical–physical characteristics of the microenvironment. Unlike traditional pipettes that measure volumes in microliters at most, microfluidics can monitor dosages in the nanoliter to femtoliter range. This technology can effectively regulate glucose [210] and oxygen levels [211], enabling us to understand the biological reactions of individual cells to changes in these levels. Microfluidics can also precisely regulate the placement of cells in different geometries [212] and organize them in 3D structures onto hydrogels, thus enabling the recreation of structures that are more similar to specific tissues of interest [213]. Other relevant advantages of microfluidics include parallelization that improves reproducibility [214,215], automation that minimizes human error [209,216], and excellent live cell imaging properties that are also able to track migrating cells and perform the phenotyping of resting cells [208,217].

Although the use of microfluidics offers an exciting and efficient approach, there are still some limitations regarding implementing PDMS media. Since PDMS is hydrophobic and porous, it may absorb hydrophobic molecules like lipids or other small molecules [218] coming from the culture media. To avoid this process, the culture media should be replaced often. Another limitation is the risk of precocious nutrient consumption and an early accumulation of metabolites. This issue may be avoided by the frequent addition of nutrients. Furthermore, due to its porosity and permeability to gases and fluids, PDMS may promote their evaporation in a non-adequately humid environment [219,220]. Other limitations relate to the intrinsic toxicity of non-properly cured PDMS [220] and cell attachment issues like channel congestion and valve malfunction [208]. Lastly, PDMS may cause an alteration in cellular growth [221] due to PDMS uptake when cells are cultivated in it for an extended period [222].

Due to their adaptable design and easy production processes, microfluidic devices are versatile tools in glioma research. They can be used for various applications, such as studying cell migration, assessing biomarkers, sorting cells from tissue samples, and testing the efficacy of treatments [223]. Device design can be tailored to the experimental goals, considering the cells’ shape, size, and density [224]. 

### 5.1. Using Microfluidics to Isolate Circulating Tumor Cells

Microfluidic devices can perform magnetically triggered cell sorting, cellular biophysics-based separation, cell-affinity chromatographic separation, and blood component separation. In the case of microfluidic cancer cell sorting, cell size differences can be exploited without requiring additional measures for biochemical characteristics [225,226]. With this technology, it could be possible to study patient-specific therapeutic sensitivity and perform gene analysis within a microfluidic chip [227]. GBM patients’ circulating tumor cell (CTC) levels have not been extensively studied compared to other types of cancer due to the brain’s unique microenvironment that prevents glioma cells from migrating into the bloodstream and the absence of extracranial metastases [228]. Nonetheless, CTCs were detected in over 33% of patient samples, indicating their presence in GBM. This finding constitutes an exciting point to explore in the research of CTC enriching techniques and noninvasive techniques to characterize the tumor treatment response [228].

With the use of microfluidic techniques that sort cells based on their affinity interactions, Wan et al. have shown that aptamers can target EGFR mutations and enrich primary tumor cells from patients’ blood. However, the success of CTC studies using patient-derived samples has been limited due to the insufficient expression of cell surface markers required for separation using these devices [229]. Newer microfluidic devices isolate GBM CTCs by excluding RBCs and platelets based on size. Inertial flow dynamics sort magnetically tagged leukocytes, leaving untagged CTCs in the solution for further processing or a cell culture [228]. In a recent study, Zhang et al. developed a microfluidic chip to quantify circulating GBM RNAs in extracellular vesicles (EV) by exploiting a transducer that recognizes target RNAs and displaces the catalytic complex. Once displaced, the catalytic portion creates a chemiluminescent reaction. The use of microfluidic chips allows for identifying individual reactions, thus enabling the detection of complex signatures of GBM sub-types directly by sampling blood [230]. Therefore, comparing the expression profiles of circulating cells to those of tumor tissue can help monitor the tumor response noninvasively during treatment and provide insights into GBM invasion [231]. 

### 5.2. Microfluidics in Molecular Diagnostics

Research on GBM requires analyzing cell-to-cell changes. Hence, combining microfluidics, cell separation procedures, and single-cell imaging can improve cancer screening and diagnosis [232,233]. The use of these techniques enables single-cell analysis with low reagent utilization, high-throughput screening, and reduced sample volumes [234]. Therefore, a new avenue for the utilization of microfluidics is its incorporation into the realm of single-cell proteomics, in which it can enable the analysis of serum or blood samples from GBM patients to pinpoint candidate biomarkers and indicators for tumor growth and response to treatment. A particular advantage for the use of microfluidic platforms is their ability to characterize the translational and signaling profiles using a limited number of patient-derived cells [235,236]. 

With microfluidics, it is possible to quantify EGFR, PTEN, phosphorylated protein kinase B (pAKT), and pS6 expression from individual cells in brain tumor biopsies, allowing for the assessment of heterogeneity. This method has been effective in analyzing single cells [236]. Recently, a microfluidic platform that utilizes immunoaffinity-based techniques to effectively separate and analyze tumor-secreted EVs was successfully developed, resulting in q high yield and applicability for molecular analysis [237]. Microfluidic technologies can combine size- and immunological affinity-based separation to rapidly and accurately identify glioma-secreted EVs. These EVs can then be examined for the transcriptional profiling of important biomarkers such as MGMT, which can help track the response to treatment [235,238]. New two-step microfluidic systems enable the more rapid extraction of EVs from liquid biopsies for mutation identification and protein expression analysis compared to traditional histopathological staining [238]. The technology is cheaper and quicker than traditional tissue biopsies, and its integration in clinical settings can facilitate the detection and characterization of patient-specific mutations and prognostic indicators [231]. 

### 5.3. Efficacy Screening with Microfluidics

The extensive migration of single GBM cells poses a substantial limitation to research in conventional cell culture methods [239]. However, studying cancer cells in 3D culture systems, such as microfluidic devices, can provide more accurate evaluations of drug metabolism, penetration, and elimination compared to traditional 2D systems [150,239].

New microfluidic systems can better recapitulate the TME and exploit interstitial perfusion to study the activity of chemotherapeutic agents on samples derived from patients’ biopsies in real time [240]. In 2020, Olubajo et al. demonstrated that GBM tissue from patients can be maintained in microfluidic chips to replicate the tumor function and tissue architecture [241]. Furthermore, other compartmentalized or multichannel microfluidic systems equipped with regulators of flow and gradient could improve the analysis of tumor cell migration, recurrence, and metastasis processes and drug efficacy and penetration into the tumor [242], also integrating microenvironment-specific chemokine gradients into the device [243]. In a study by Ozturk et al., a microfluidic device was constructed using extrusion-based bioprinting to monitor and evaluate the reactions of GBM cells to TMZ treatment, with a patient-derived GBM tumor spheroid placed between two perfused vascular channels. They demonstrated that even with long-term TMZ exposure, some GBM cells retain their invasiveness [244]. A study by Akay et al. has also used a microfluidic organ-on-a-chip to test the effectiveness of TMZ-based combination therapy by comparing the results of TMZ alone, bevacizumab alone, and a combination of TMZ and bevacizumab. Their results highlighted an overall better effectiveness with combined TMZ and bevacizumab compared to monotherapy TMZ treatment, which was, however, still superior to monotherapy bevacizumab treatment [245]. Moreover, a study by Nizar et al. explored the utilization of propofol to induce cell death in GSC niches. They used microfluidic platforms to assess the survival of GSCs after propofol administration. After the GSC spheroids were dissociated, they were cultured in chip chambers and exposed to different concentrations of propofol, and survival was monitored by continuous fluorescent microscopy. Microfluidic assays demonstrated that propofol caused a significant decrease in cell survival, selectively targeting tumoral cells [246]. 

Even without needing patient-derived tissues, microfluidic devices can quickly introduce drug resistance in GBM cell lines to study resistance development [231,247,248]. Using microfluidic platforms in experiments has several advantages, including incorporating 3D culture systems for more accurate studies and creating therapy-resistant cell lines. Indeed, building a microfluidics system that concomitantly contains tumor spheroids and a monolayer of endothelial cells scaffolded in a 3D hydrogel allows for a better inhibition of the epithelial–mesenchymal transition and improves antimetastatic drug responses. Notably, in a study comparing the 2D and 3D models, there were dramatic variations in the responses to therapeutic agents [249].

BBB models recreated via microfluidic devices have accurately replicated preclinical drug delivery into the tumor mass [250,251,252]. As confirmed by biochemical analyses, incorporating endothelial cells into these models has produced tight junctions that more closely mimic BBB penetration [251]. Later versions of these microfluidic devices have added astrocyte/endothelial cell cultures and fluid shear stress to enhance preclinical testing for BBB drug permeability screening [250,252]. In this context, Marino et al. presented a 3D microfluidic BBB model to study drug diffusion. Their system is equipped with porous microtubes to model microcapillaries and endothelial cells surrounding them. With this model, they studied the penetration and efficacy of nutilin-3a (nut-3a), demonstrating its efficacy in promoting glioma cell death when administered in adequate concentrations [253]. Microfluidics may also address the difficulty in employing nanoparticles in trials studying drug carriers. Indeed, microfluidic devices could be helpful in the synthesis of drug-loaded nanoparticles. They are also helpful in performing drug distribution and targeting assays by strictly controlling reagent mixing temperatures and timing and managing the distribution and loading of the drugs within nanoparticles [254]. A study by Mendanha et al. used a microfluidic system to explore the use of docosahexaenoic acid liposomes as an inhibitor of GBM migration and proliferation and a promoter of apoptosis and autophagy [255]. The use of microfluidics was advantageous, as it allowed for monitoring and adjusting flow rates, flow rate ratios, and lipid concentrations, thus allowing for the fine-tuning the physicochemical characteristics of the liposomes. The group examined how different sizes of liposomes are internalized by GBM cells and identified the sizes with the highest success of internalization. They also confirmed that docosahexaenoic acid can induce apoptosis in GBM cells through a variety of mechanisms including PARP cleavage and pro-caspase 3 activation [255].

### 5.4. Microfluidics in Studying GBM Progression and Cell Localization

Changes in the cellular phenotype relate to the genetic factors that influence tumor heterogeneity, chemoresistance, and cell proliferation [256]. In order to provide a clear insight into understanding particular mechanisms of tumor advancement, it is possible to create microfluidic devices with microchannels that can simulate how GBM cells migrate throughout the interstitial regions of the brain [242].

A novel technique called “organ-on-a-chip” has been devised by combining the principles of tissue engineering and microfluidics to create artificial systems capable of mimicking the human body’s intricate physiological processes and organ interactions [231,248]. An organ-on-a-chip GBM model devised by Yi et al. demonstrated patient-specific and clinically matched responses to drug treatment combinations, including chemoradiation combined with TMZ [257]. Relying on these models, other groups also suggested that proneural cells are typically located around the blood vessels and have lower mortality, whereas mesenchymal-type cells have higher invasiveness [193].

In conclusion, models of cancer-on-a-chip can mimic complex 3D microarchitectures at the organ level accurately and in real time [231]. They can also quantitatively measure cellular responses and assess invasion, migration [258], and angiogenesis [259]. Additionally, they are capable of characterizing intravasation and extravasation [260].

### 5.5. Microfluidics for Studying Extracellular Matrix Signaling

The ECM significantly controls invasion in GBM cells. Signals from these cells cause dynamic remodeling of the ECM to promote invasion [261]. Individual GBM cells concentrate integrins, thus producing adhesion forces at the focal contacts with ECM [262]. Invading cells release matrix metalloproteases (MMPs) to alter the local ECM during mesenchymal migration [262,263]. Unfortunately, therapeutic approaches targeting MMP-dependent migration have not successfully treated cancer metastasis. Microfluidic devices could help to examine physical interactions with the ECM to understand how tumor cells modify their shape and spread [264,265]. Hydrogels with MMP-degradable sites can also be utilized to examine actual ECM remodeling by tumor cells in vitro, further improving these platforms [231,266]. 

### 5.6. Microfluidic Devices in the Study of Cell–Cell Interactions

Microfluidic devices can assist in understanding the self-organization of GBM cells and how paired cell interactions impact the cellular architecture of tumors [231,267]. Using microfluidics as a research tool enables the investigation of the critical interplay between tumor cells and endothelial cells, which contribute to the advancement of cancer and the presence of CTCs [268]. Specifically, the process of GBM cell migration through endothelial barriers, which is difficult to assess via traditional 2D cell culture systems, is a pivotal component of the invasion cascade. The study of these mechanisms benefits from devices that can change and examine the components of the stroma—for example, to study the role of gap junctions between cells [269]. 

Another interesting interaction to study is the one of GBM with microglia. Microglia can either hamper the initial growth of the tumor by phagocytosis or promote invasion and cell proliferation [270]. Guo et al. explored this bidirectional relationship between GBM and microglia by designing three microfluidic co-migration assays [271]. To further characterize this interaction, another study performed by Hong et al. explored the inhibitory effects of micro-RNA (miRNA)-124 EV on GBM cells. These miRNA EVs appeared to have anti-proliferative activity, and they could also inhibit the M2 polarization of the microglia [272]. Another niche that has been studied using microfluidic assays is the perivascular niche (PVN). Adhei-Sowah et al. designed an organotypic tricultural microfluidic model to model the interaction between the PVN and GSCs and how the PVN may in-fluence GSC invasion, proliferation, and stemness. Their model included astrocytes, endothelial cells, and GSCs. Their model was positive for stemness markers such as CD44, Nestin, and SOX2, indicating a successful recapitulation of the stem cell niche. The results highlighted the significant contribution of astrocytes in stimulating aggressiveness in GSCs within the PVN. They were also able to identify 15 ligand–receptor pairs that may contribute to GSC migration by means of chemotactic mechanisms. In particular, they demonstrated that the SAA1-FPR1 ligand–receptor pair can drive GSCs towards the PVN and promote invasion [273].

### 5.7. Microfluidics for Modeling Interactions with Vascular Flow, Hypoxia, and Angiogenesis

The buildup of a pressure gradient around the edges of brain tumors can cause interstitial fluid to flow through the brain tissue, which may lead to the outward migration of tumor cells [274]. Additionally, the collapse of the surrounding blood vessels and increased fluid flow can cause the cells within the tumor to experience greater levels of fluid shear stress [275]. To recreate an in vivo-like vasculature, microfluidics devices exploit collagen and stromal cells to create microchannels that simulate crucial BBB and TME characteristics [276,277].

Microvascular hyperplasia, a characteristic of GBM, is thought to be related to the distribution of pseudopalisading cells at the periphery of the tumor, promoting cell migration [278]. By manipulating flow rates and the resulting shear stresses, microfluidic devices can assess vascular perfusion [277,279]. Moreover, microfluidic devices with perfusable microvascular networks also enable researchers to acquire images showing how tumor cells interact with the flux of fluids in real time, thus giving further insight into how tumor cells take advantage of the interstitial fluid dynamics to migrate and metastasize. These techniques also allow for more realistic simulations of in vivo conditions [280]. In addition, microfluidic devices can mimic the process of angiogenic budding at the cellular level, including the hypoxic response, and offer control over the conditions that regulate oxygen tension [281]. Specifically, to simulate a hypoxic environment in 3D models, the design of microfluidic devices involves the placement of gas channels both above and beneath the scaffold for the cells. In this way, the structure grants more precise control of oxygen levels in the culture [282]. To further recreate this environment, polycarbonate films are positioned above the cell channels to lower oxygen diffusion from the atmosphere, whereas PDMS allows oxygen diffusion regulated by a specific channel [283].

Cui et al. designed a microfluidic and biomimetic model to study angiogenesis in GBM. According to their model, angiogenesis may be regulated by TGFβ1 and interactions between the surface endothelial cells and macrophages. At the perivascular level, angiogenesis was shown to be promoted by macrophage–endothelial interactions through integrin alpha-v-beta-3 [284].

### 5.8. Microfluidics for Modeling Immune Cell Interactions

Microfluidic devices can be used to study the immune–tumor cell interplay. In this context, studies have highlighted that it is necessary to evaluate more than one cell type at the same time to fully grasp the importance of cell interactions in the development of metastases as the immune system continuously changes throughout the progression process [285,286]. Unlike conventional in vitro cultures, microfluidics and 3D models can consider the dynamicity of cancer-targeting immune cells in space and time. Research seeking to explore the interactions between glioma and the immune system can benefit from organ-on-a-chip models [287,288]. Through microfluidic tests conducted in these models, it is possible to evaluate different therapeutic combinations based on the genetic landscape of GBM biopsy samples from patients and assess the effectiveness of future immunotherapies. These platforms also help to determine the degree to which immunotherapies boost the body’s natural defenses against cancer [231].

The GBM-on-a-chip model by Cui et al. suggests that it is possible to use targeted immunotherapy to block both alpha-v-beta-3 integrin and TGFβ1 at the same time in order to decrease neovascularization. In this way, it is possible to tackle both macrophage–endothelial cell interactions and macrophage-associated immunosuppression [284]. Microfluidics can also be employed to study and improve the mechanisms of cell-based immunotherapy. For example, a study by Huang et al. involved a microfluidic device to study BBB penetration by CAR T cells [289]. The study’s results indicate that the microfluidic platform has the potential to comprehend the effects of CAR T therapy on cells located beyond the BBB, also considering concurrent toxicity and the processes of BBB breakdown [290].

## 6. Animal Models Used for In Vivo GBM Research

Over the last 60 years, animal models for primary brain cancers have experienced continuous evolution, and considerable advancements have been achieved, most recently with the introduction of highly invasive GBM models [15]. The latter models have provided crucial insight into specific mechanisms underlying the development of brain tumors. 

Since patient tissue was first transplanted into rats in the 1940s [291], various cancer models, including those simulating gliomas, have been successfully established. This, combined with the development of chemical in vivo carcinogenesis, gave researchers a greater knowledge of the molecular and cellular pathways underpinning tumor growth and heterogeneity [292]. As in vivo models replicate tumor behavior in a whole mammalian organism and reflect actual disease, they (1) enable a better understanding of tumor growth and biology, (2) validate the efficacy and safety of current medicines, and (3) aid in the discovery of novel therapeutic approaches. Additionally, in vivo models mimic important features of carcinogenesis such as the biological microenvironment, angiogenesis, and immunological and inflammatory responses, which offers distinctive and valuable perspectives into brain tumor biology [15]. Animal models have therefore proven to be essential in understanding the biology of GBM and assessing prospective treatment approaches.

Mice have been the main species used in preclinical animal cancer modeling, and their adaptability for future therapy development has, in part, contributed to better outcomes for cancer patients. As a result, many rodent-based models of brain cancers have been created, including xenograft models, genetically engineered mouse models, and carcinogen-induced rodent models [293]. However, with the translation of these preclinical results into pharmaceutical development, the shortcomings of murine cancer models have prompted researchers to look for substitute animal models, including large animal models [294]. This is because mouse models of human disease face a number of anatomical and physiological constraints. Porcine, canine, and, less frequently, non-human primate (NHP) models are examples of potential large animal models. In this section, we will review the different types of in vivo animal models used in GBM research (summarized in Figure 2).

### 6.1. Murine Models

Since the mid-1970s, rat models have been relied on extensively in preclinical neuro-oncology research. Although about 1% of the common laboratory rat strains develop brain tumors on their own [295], experimental cancers can be generated in rodents by administering alkylating substances like N-methylnitrosourea (MNU) or N-ethyl-N-nitrosourea (ENU) [292]. These agents can be delivered orally, intravenously, or via transplacental injection. The earliest GBM models were created by injecting pregnant rats and mice with nitrosourea in order to generate brain tumors in their offspring [296]. For instance, the transplacental injection of the carcinogen ENU causes the progeny to develop a variety of CNS cancers, including tumors that resemble gliomas [297]. It is significant to keep in mind that the timing of transplacental carcinogen exposure is crucial, with the best results being shown 18 days after conception [298]. Some of the chemically induced rat brain tumor models include the previously discussed syngeneic 9L gliosarcoma, C6 glioma, and F98 glioma [292,299].

Rat brain tumor models enabled the generation of tumors de novo all while maintaining tumor–host interactions, thus providing heavily beneficial data for the development of several innovative treatment options for human high-grade gliomas. Moreover, mouse models attracted interest since they can be genetically altered in numerous ways in order to assess the effects of various mutations on tumorigenesis and tumor sensitivity to various interventions [293,300]. Rodent models have gained significant prominence in oncological research due to several compelling reasons. First, their relatively small size offers an optimal balance between experimental utility and resource efficiency and is conducive to cost-effective care. Second, their rapid reproductive rate proves essential for sustained, long-term experimental maintenance. Third, the thorough characterization of the mouse genome has made genetic modifications both simple and precise, leveraging contemporary technology to generate murine tumors that closely resemble their human counterparts [301]. Consequently, mice have emerged as the organism of choice for simulating the intricate genetic and physiological aspects of cancer. This has been reflected by the expansion of mouse glioma models and the productivity of preclinical investigations which advanced our comprehension of the molecular characteristics of GBM [300]. Nevertheless, the clinical translation of these advancements continues to face challenges, as indicated by the limited success rates observed in clinical studies. 

Significant differences between rodents and humans largely limit these models’ translational capacities, as none of the current models exactly replicate human GBM [302]. The mouse brain is much smaller, lacks the gyration and cortical development characteristic of the human brain, and fails to model neural network phenomena [303]. These anatomical and functional differences limit the replicability of human tumors in rodent models in which tumors are circumscribed as compared to the infiltrative and invasive nature of high-grade human gliomas [20]. Moreover, it is important to note that chemically induced CNS carcinogenesis appears to be species-dependent, as it has exhibited wide success in a myriad of rat strains, yet a similar success rate has not been observed in mice [304]. Furthermore, there are significant differences in the tumor histological characteristics between the various chemically engineered models, which poses an issue when attempting to study a tumor as complex as GBM [300]. It is also still uncertain to what extent the mutational and transcriptional patterns of rat cancers resemble those of human malignancies since rat tumors have not been thoroughly described at the molecular level and human tumors are heterogeneic. Therefore, although numerous in vivo rodent brain tumor models have been established and have been crucial for our understanding of gliomas to date, it is evident that most, if not all, fail to adequately reproduce the complex genetic and phenotypic profiles of human glioblastoma.

### 6.2. Canine Models

Canine models have shown spontaneous GBM formation, providing the opportunity to examine glioma without the use of external manipulative agents [305]. In fact, research has demonstrated that the incidence of GBM in dogs is comparable to that reported in humans [292]. Moreover, histopathological studies have additionally shown that, in comparison to murine models, gliomas detected in canines mimic those seen in humans to a considerably greater extent and exhibit similar neural precursor markers [306]. This makes canines an interesting model for testing innovative therapeutic approaches for GBM. In 2015, the Comparative Brain Tumor Consortium, established by the National Cancer Institute, was designed to direct the investigation of canine brain tumors in order to find new therapies for patients with human brain tumors [307]. However, the challenge in identifying canine brain cancers makes its implementation problematic. In fact, despite the higher likelihood of spontaneous brain tumor growth in dogs, there are still far fewer canine cases accessible for research compared to mouse models [300]. The use of canines in testing is additionally restricted by ethical constraints, and access to companion animals is challenging for adequately controlled studies. Nonetheless, because of its diverse phenotypic composition and close similarities to human cancer pathophysiology, the canine model has tremendous potential for human cancer research.

### 6.3. Non-Human Primates

Non-human primate (NHP) models fill the gap between rodent models and humans. As NHPs offer significant advantages over other models in terms of anatomy, physiology, and neurobiology, they are valuable in the study of human disease, as they may best recapitulate human tumor behavior as well as enable the evaluation of sophisticated therapeutic approaches such as surgical procedures, adjuvant therapy, and imaging techniques [294,308]. However, cancer studies in NHPs have faced substantial limitations, stemming from ethical concerns, extensive costs, and practical difficulties. For instance, the ongoing lack of specialized laboratory equipment required for the effective management and genetic modification of large animal models restricts the feasibility of inducing tumors in NHPs [292,309]. Nonetheless, NHP models may be used as a transitional stage in the development of experiments that more accurately mimic human physiology and hence increase the success rate of clinical trials. It is anticipated that increased interest in the field will increase their relevance and utility [310]. 

### 6.4. Other Animals

One of the most sophisticated and adaptable large-animal glioma model is the porcine, which has a long history in biomedical research [311]. The pig brain closely resembles the human cortex anatomically and can reproduce drug administration, drug diffusion, and tumor infiltration within cortical regions [312]. Other advantages include a bigger brain size, which allows for high-resolution imaging and pharmacokinetic distribution studies, a high litter capacity, with up to 20 offspring a year, and fewer ethical limitations [313]. Their application in preclinical investigations is also supported by the recent development of numerous porcine glioma models [294].

Zebrafish and drosophila are two additional non-murine animal models used in brain tumor research. These models are excellent for the development of brain tumor models because they exhibit high levels of evolutionary conservation and physiological similarities to humans, including possessing the BBB [314]. Several advantages make these models perfect for neuro-oncology research. For instance, zebrafish embryos are transparent, which facilitates tumor monitoring. They also offer several ethical and financial advantages and are characterized by their simplicity of genetic manipulation [315]. In this regard, new insights into the pathogenesis of GBM have been successfully generated using genetic and xenotransplant zebrafish models [316]. Nonetheless, more research is needed to generate other specific glioma-resembling tumors in these animals.

## 7. Generation and Applications of Available Animal Models

### 7.1. Grafting Tumor-Initiating Cells

There are three main questions that define the process of grafting tumor-initiating cells into animal models: What is the origin of the grafted cells? What is the immune status of the recipient animal? Where are the cells being implanted? The answer to the first question determines whether the graft is an allograft, meaning that it is coming from an origin of the same species as the recipient animal, or, alternatively, a xenograft, meaning that it is coming from a different species, typically humans in this case. In fact, the type of the graft also dictates the answer to the second question regarding the immune status of the recipient. In this context, syngeneic transplantation refers to the use of allografts from carcinogen-induced tumors or established cell lines and transplanting them into genetically matched hosts (Figure 3). This type of transplantation carries the major advantage of having an immunocompetent host, which allows for the testing of immunotherapies and characterizing the interaction between the immune system and GBM [317]. The most common examples of syngeneic transplant models include the 9L, RG2, and F98 cell lines that are usually implanted in Fisher rats, the CNS1 cell line implanted in Lewis rats, and the GL261 and CT2A cell lines implanted in C57BL6 mice [318]. These cell line–animal model complements have been extensively used in GBM research and have contributed greatly to the knowledge we have regarding the immune profile of this tumor [319,320,321]. Notwithstanding this, there are several limitations and shortcomings for these models. Notably, as with other immortalized cell lines that are repeatedly propagated, these lines are subject to genetic drift and a subsequent loss of the authentic recapitulation of GBM features [26]. Moreover, the fact that these lines are generated from murine models of GBM and are implanted in a host with a murine immune system undermines the generalizability of research findings to human tumors and the ability to replicate these findings in clinical trials [294].

Therefore, another option available is the use of established human cell lines, which were originally generated from human GBM tumors and subsequently propagated in laboratories. The earliest and most popular among these lines are the previously discussed U87 and U251 cell lines. When these cell lines are transplanted into animal models, they are considered xenografts, and thus, the hosts are required to be immunocompromised for the tumors to grow (Figure 3). The development of immune-depleted mice contributed significantly to the possibility of such engraftments, with several available options and degrees of immunocompromised variants. These include nude athymic mice that cannot develop mature T-cells, non-diabetic obese mice with severe combined immunodeficiency (NOD-SCID), SCID-beige mice that lack both functional B- and T-cells, and NOD-SCID mice with a further deletion of the interleukin 2 receptor gamma chain (NSG) or the deletion of JAK3 (NOJ), which are devoid of B-, T-, and Natural Killer (NK) cells [322,323]. Interestingly, zebrafish offer the ability to transplant xenografts without the need for the genetic engineering and breeding of specialized immunocompromised strains, like those discussed in mice. This is because zebrafish pass through a natural immunocompromised state that is inherent to their development. Specifically, the adaptive immune system is deficient in zebrafish during the first 12–14 days post-fertilization, thus offering a period that can be utilized for implanting established cell lines of human GBM [316]. Despite the fact that the mentioned cell lines are originally derived from human tumors, they are also subject to the risk of genetic drifting and divergence from the morphologic and histologic characteristics of the initial tumor. Moreover, a major limitation of these cell lines is that they fail to mirror the intratumoral heterogeneity that is present within human GBM [318]. Various cell subpopulations were discovered within GBM tumors, with each subpopulation having a distinct genetic and molecular profile and a differential response to therapies [324]. The available established lines are not representative of this heterogeneity. Hence, a new research avenue has gained significant traction, using PDXs. This model allows for the transplantation of cells or spheroids that are directly derived from human GBM tumors into immunodeficient animals, without the need for an intermediate step of passaging via cell cultures (Figure 3). Consequently, the resulting tumors can more accurately encompass the diversity of cell subpopulations and reproduce the complex stroma of human GBM [325]. The superiority of this model has been proven through comparing the characteristics and response to treatment of the grafted tumors in animal models to the original tumors they were derived from in patients. Indeed, the tumor models resembled their parent tumors in their invasiveness, immunohistochemical profile, radiation sensitivity, and response to TMZ [326]. Additionally, one specific challenge was the response to antiangiogenic therapy with bevacizumab. In this context, previous experiments using the U87 cell line showed a survival advantage for mouse models after treatment with bevacizumab, while clinical trials failed to show a similar effect for the drug in humans. Intriguingly, PDX-based models also failed to show a survival advantage after treatment with bevacizumab, just like their human counterparts. This indicates that the use of PDX models can better predict the results of clinical trials and determine which candidate drugs should be investigated in humans [326]. However, the limitations of this model are related to the requirements of continuous in vivo passaging, which incurs a greater financial burden on laboratories and is more cumbersome for research personnel. Additionally, even though these grafts are passaged in vivo, this does not protect them from the inherent risk of genetic drifting and the gradual replacement of the native TME with a murine one [327,328]. Finally, just like other xenografts, PDXs require the presence of an immunocompromised host. It is worth mentioning that a more personalized model of PDX is gaining interest in oncology research, and it includes the creation of “Avatar” mouse models for patients. This can help with the more specific profiling of a patient’s tumor characteristics and offer more reliable information regarding the response to treatment regimens and certain targeted therapies [329]. However, there is a debate regarding the practicality and efficiency of such models, since they require more time for development before indicating any clinically meaningful information, which may not be suitable for patients with advanced tumors [318].

One common shortcoming for all of the mentioned models is their inability to provide adequate characterization of the interaction between human GBM and the human immune system and properly test the potential for immunotherapeutic interventions. Based on that, several efforts have been directed towards deriving a model that can overcome this challenge. Herein, humanized mouse models were developed to provide a possible avenue. These models include the use of immunocompromised mice with a transplanted HLA-matched human immune system. This was first achieved by engrafting human peripheral blood mononuclear cells (PBMCs), which were able to generate functional human T-cells but not fully functional B-cells. However, this model developed graft versus host disease (GVHD) within 4–6 weeks [330]. An upgraded model involves the transplantation of human CD34+ hematopoietic stem cells (HSCs), which were able to generate a wide array of leukocytes, including T-cells, B-cells, and NK cells. Moreover, it offered a longer time period (10–12 weeks) before the development of GVHD. Nevertheless, the generated T-cells are dysplastic and do not mature properly [330]. Consequently, researchers developed a model with an implanted human fetal thymus and fetal liver, in addition to the injection of HSCs, to ensure the adequate maturation of T-cells and other immune components. Indeed, the bone marrow, liver, and thymus (BLT) model ensures better development of T-, B-, NK, and myeloid lineage cells [330]. However, GVHD might still develop within 20 weeks. Several techniques have been studied to further improve the outcomes of these models, with many effective amendments. These include the injection of human cytokines and immune mediators, such as interleukin-7 (IL7) and granulocyte colony stimulating factor (G-CSF), and the complete suppression of the host’s autoimmune system [330]. The advent of humanized mouse models provided a suitable niche for testing the efficacy of immunomodulation agents such as anti-PD1 in the setting of GBM [331]. Nevertheless, the development of such models is laborious and expensive and thus would limit the accessibility and utility of this model. Hence, several efforts were directed towards the creation of a model that allows for the grafting of human GBM into animal models with a competent immune system. Here, the recently developed “immunotolerant mouse model” comes as a promising avenue that makes this process possible. This model was achieved through a T-cell pharmacotherapeutic blockade using abatacept, which is an immunoglobulin that mimics the role of CTLA-4 by binding to CD80/86 and preventing the subsequent activation of T-cells, and MR1, an anti-CD154 antibody that prevents the stimulatory interaction between CD154 on T-cells and CD40 on antigen-presenting cells. These agents were delivered intraperitoneally to immunocompetent C57BL6 on days 0, 2, 4, and 6 after the grafting of human GBM cells [28]. The resulting models successfully grew intracranial tumors with preserved interactions with the host immune system, which closely recapitulated the pathophysiological development and characteristics of human GBM [28]. In the future, the immunotolerant mouse model may serve as a simple and cost-effective method that can further accelerate research in the field of GBM immunotherapy. Moreover, it may be even generalized to create similar immunotolerant rat, porcine, and canine models. 

Finally, the answer to the third question determines whether the graft is orthotopic, meaning that it is implanted into the brain, or heterotopic, meaning that it is implanted into a different organ, mainly the subcutaneous fat in the flank area. Expectedly, the orthotopic option carries several advantages pertaining to the recapitulation of the constraints of the BBB and intracranial immune privilege [317]. Moreover, orthotopic grafts have been shown to more faithfully preserve the histological and molecular landscape of human GBMs [326,332]. On the other hand, heterotopic grafts carry advantages related to practicality, such as the ease of observing tumor growth and avoiding the unexpected death of animal models due to the unclear tumor burden, which can occur with orthotopic transplants [318].

### 7.2. Engineering Models with Spontaneously Arising Tumors

The identification of the commonly deleted and overexpressed genes in GBM tumors paved the way for the engineering of animal models that can spontaneously develop such tumors on their own. These models are generated by genetic engineering or the viral vector-mediated introduction of genetic alterations (Figure 3). The results mainly involve gain of function mutations in oncogenes, such as *KRAS*, *BRAF*, *EGFR*, *PIK3CA*, or *CDK4*, or loss of function mutations/the deletion of tumor suppressor genes, such as *TP53*, *PTEN*, *RB*, or *CDKN2A* [333]. The advantage that this model has over the grafting models is that it recapitulates the tumorigenesis process and offers an overview of GBM development from its early stages through the later ones, with the ability to test therapeutic interventions at each stage. Moreover, the genetically engineered models do not involve invasive procedures for achieving tumor implantation. Consequently, the BBB and general brain architecture remain unaltered, and the mice are not exposed to the additional burden of surgical procedures [318]. In addition, the animals have a fully functional immune system and allow for the studying of tumor–immune system interactions. However, these models carry a similar limitation to that of syngeneic grafted mice, which is that although the tumor carries similar genetic and molecular characteristics as human GBM, they fail to copy the human TME and immune system [318]. Herein, we will elaborate on the various methods that have been employed to create animal models that spontaneously develop GBM tumors.

#### 7.2.1. Genetically Engineered Models

Genetic engineering refers to the manipulation of the genome of zygotes or embryos or the breeding of certain strains to achieve the insertion, deletion, or alteration of certain genes. For instance, the Cre-LoxP system entails crossing a Cre animal model which expresses the Cre recombinase with a Lox animal model, which has a target gene or cassette surrounded by two LoxP sites. Breeding these two strains results in a model with a complete Cre-LoxP system, in which the Cre recombinase can cut out the sequence between the two LoxP sites, which will result in the deletion or activation (if the removed sequence was a “Stop” cassette) of certain genes [334]. Importantly, the Cre-LoxP system offers spatiotemporal control over the genetic recombination process. Spatial control is ensured by linking the expression of the Cre recombinase to that of an organ/tissue-specific promoter, such as Nestin (NES) for the central nervous system [335]. Temporal control is attained by fusing the Cre protein to another regulator that can act as an on/off switch for the Cre-LoxP system. For instance, a commonly used regulator is the estrogen receptor (ER) with a mutated ligand binding domain, which is fused to the Cre recombinase. In its initial state, the Cre-ER complex binds the heat shock protein 90 (HSP90), which prevents its nuclear translocation. Once the system needs to be activated, tamoxifen is administered systemically, and it disturbs the binding of Cre-ER to HSP90 and allows the Cre-LoxP system to execute its function [335]. A similar genetic recombination system can be constructed using the FLP recombinase and the FRT flanking sequences. This system can also be preferentially expressed in certain tissues [336]. For instance, coupling FLP to the promoter of the GFAP ensures that this system is spatially confined to astrocytes and neural stem cells that express GFAP. Furthermore, using the Cre-LoxP and FLP-FRT systems concomitantly was proven to be feasible as a potential avenue for designing better GBM models and offering researchers greater control over the genetic alterations in the generated models [336].

Another system that has been utilized for the creation of animal models that spontaneously develop GBM is the transposon system. Transposons are DNA sequences that have the ability to change their location within the genome using a “cut–paste” mechanism named transposition with the help of the transposase enzyme. The DNA sequences are marked by terminal inverted repeats (TIRs) that surround them on both sides [337]. These elements have been studied and exploited as a mechanism for genetic engineering, and this transposition process can now be applied to any DNA sequence of interest if it was inserted between two TIRs and the transposase enzyme was supplemented. In this context, the most commonly used transposon systems that have been used to generate GBM models are the Sleeping Beauty (SB) system and the Piggyback system [15]. For instance, the SB system was successfully used to create the SB28 spontaneous GBM model by inserting the genes *NRAS*, *PDGF*, and short hairpin *TP53* (which can silence the *TP53* gene) [338,339].

Finally, a more cost-effective and simple method that was more recently developed is the CRISPR-Cas9 system, which can be used to knock out specific DNA sequences. The system functions as an RNA-guided nuclease that can cut out specific sequences that are predetermined by single guide RNA (sgRNA). Contextually, a GBM mouse model was derived using the CRISPR-Cas9-mediated knock out of *PTEN*, *TP53*, and *NF1* [15].

#### 7.2.2. Viral Vector-Induced Models

Viral vectors can be used to transduce specific somatic genetic changes to adult animal models. These vectors can be administered through several different routes, each having advantages and disadvantages. For instance, intracranial and intraventricular delivery routes can achieve the localized delivery of the viral vectors and require a lower viral load to achieve their effects; however, these techniques involve breaching the BBB and exposing the animals to the stress of stereotactic injection. On the other hand, viral vector delivery can be carried out through the intrathecal or intravenous routes, which are less invasive but necessitate the delivery of large quantities of the virus to obtain satisfactory results, thus risking systemic toxicity or reactions [340]. Similarly, several types of viruses have been exploited for their ability to serve the purpose of delivering DNA cargo to specified tissues/cells. 

One of the most commonly used viral vector-mediated systems relies on retroviruses, mainly the avian leukosis sarcoma virus splice-acceptor system (RCAS). This system can successfully transduce replicating cells that express the tumor virus A (TVA) receptor. The expression of this receptor can be linked to that of certain tissue-specific genes, such as *GFAP* and *NES*, to achieve the spatial control of the transduction process. For instance, Hambardzumyan et al. used the RCAS system to deliver the *PDGFB* gene to mice with GFAP-linked TVA, which resulted in the development of low-grade gliomas in wild type mice. However, once introduced into mice carrying loss of function mutations in the *TP53* or *ARF* genes, the PDGFB-RCAS system resulted in the development of GBMs in approximately 100% of the transduced mice [341]. Other utilized RCAS systems involve the introduction of *EGFRvIII*, *AKT*, or *HRAS* [340]. One advantage of the RCAS system is that the used virus does not replicate in mammalian cells and thus does not lead to altered characteristics of the generated tumor cells due to the burden of viral propagation. However, the shortcomings of this system are the limited size of the cargo that it can deliver (around 2.5 Kb) and its ability to target mitotically active cells only. Hence, two other viruses that have gained attention as potential vectors for the transduction of animal models are adenovirus and lentivirus, which can carry larger cargo (>10 kb) and possess the ability to target dividing and non-dividing cells [342]. Specifically, a combination of the Cre-LoxP system and lentivirus delivery system has been utilized to induce the expression of *HRAS* and *AKT* genes and resulted in the development of GBMs in adult mice models [343]. In a parallel fashion, the adenovirus-mediated delivery of the *EGFRvIII* gene into mice with an *RAS*-activated genetic background efficiently led to the development of GBM in mouse models [344].

## 8. Conclusions and Future Perspectives

The preclinical models available for the investigation of GBM have travelled a long road of evolution and advancement to reach the current status quo (summarized in Table 2). The scientific community has been putting great efforts into understanding the genetic, epigenetic, molecular, and neurodevelopmental profiles of human GBM in order to create models that can faithfully mirror the behavior of this tumor. The recent development of humanized and immunotolerant mouse models of GBM can offer great insights into the tumor–immune system relationship and how to effectively modulate it. Moreover, the emerging fields of 3D in vitro modeling and microfluidics can revolutionize the field of GBM research, allowing for the high-throughput screening of therapeutic agents in a time-efficient and cost-effective manner. 

Nevertheless, the dramatic leaps that have been made in the development and advancement of GBM preclinical models have not been translated into real-world practice-changing results yet. Further optimization of the pipeline of preclinical research and properly utilizing the available models can enhance the translatory potential of preclinical findings. For instance, experiments should aim to investigate the superiority or added benefit for any novel therapeutic agent or treatment modality as compared to the already established standard of care. Moreover, experimental models should strive to recapitulate human GBM to the greatest extent possible by implementing the latest technologies and considering the accumulated knowledge we have on GBM evolution, invasion, and interactions. Finally, the careful and proper integration of artificial intelligence and machine learning into the field might empower the available models, complement the efforts of research personnel, and accelerate the discovery of relevant findings.

## Figures and Tables

**Figure 1 ijms-24-16316-f001:**
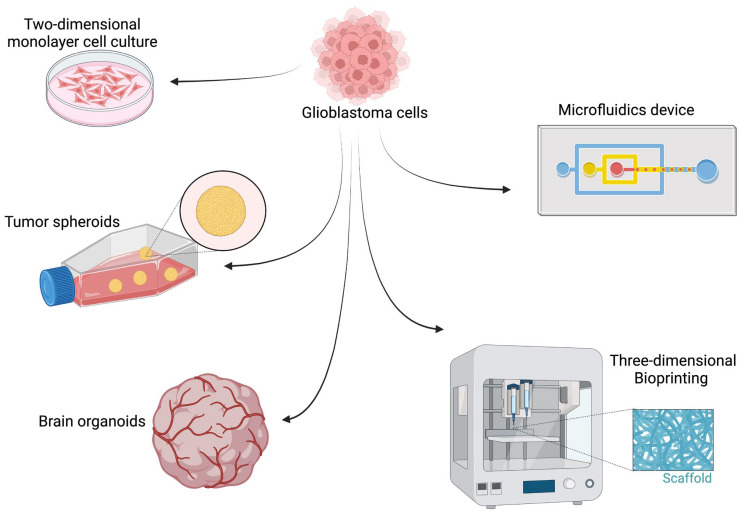
Illustration representing the different in vitro models used in glioblastoma research.

**Figure 2 ijms-24-16316-f002:**
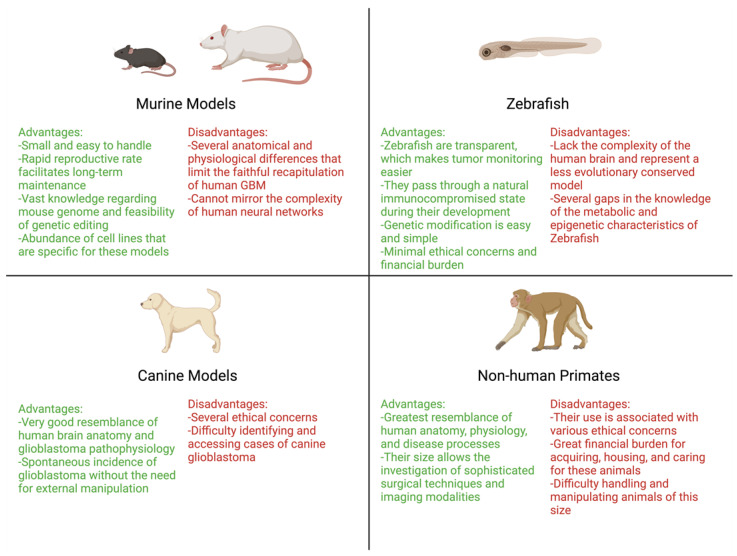
Illustration representing the different animal models used for in vivo glioblastoma research, along with the advantages and disadvantages of each.

**Figure 3 ijms-24-16316-f003:**
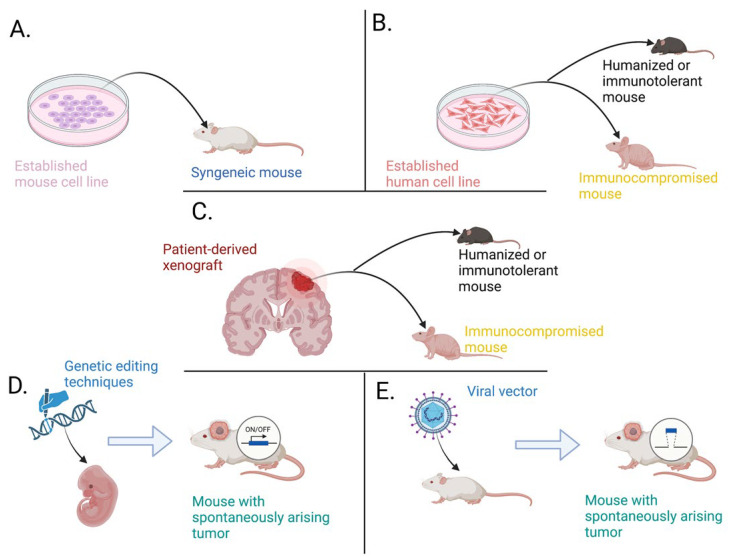
The different methods used for the generation of animal models of GBM. (**A**) The implantation of established cell lines into syngeneic animals of the same species allows for the use of immunocompetent hosts. (**B**) The use of established human cell lines that were initially obtained from patient tumor samples and propagated in culture require implantation into immunocompromised, humanized, or immunotolerant animal models. (**C**) Similarly, the use of patient-derived xenografts involves the implantation of patient tissue into immunocompromised or humanized/immunotolerant hosts; however, this type of xenografting does not include in vitro passaging as an intermediate step. (**D**) The genetic engineering of animal embryos or the breeding of genetically engineered parents can produce animal models that develop tumors on their own in a fashion that can be temporally and spatially controlled. (**E**) In a similar way, the use of viral vectors to transfect animals allows for the generation of tumors “from scratch” in a temporally and spatially controlled manner.

**Table 1 ijms-24-16316-t001:** The genetic and histopathologic characteristics of established glioma cell lines.

	C6	9L	F98	RG2	GL261	CT2A	U87	U251
Origin	Wistar Rat	Fischer Rat	Fischer Rat	Fischer Rat	C57BL/6 mice	C57BL/6 mice	Patient	Patient
*P14ARF* mut.	−	−	NA	+	+	NA	+	+
*P16* mut.	+	−	+	+	+	NA	+	+
P53 mut.	−	+	−	−	+	−	−	+
P*TEN* mut.	−	−	NA	NA	+	+	+	+
*RAS* mut./overexpression	+	NA	+	+	+	NA	+	+
*IDH1* mut.	−	NA	NA	NA	−	−	−	−
*IDH2* mut.	−	NA	NA	NA	NA	−	NA	NA
EGFR overexpression	+	+	+	−	+	NA	−	+
PDGF-β amplification	+	−	+	+	NA	NA	NA	NA
GFAP	+	+	+	NA	−	+	−	+
S100	+	+	−	NA	NA	NA	−	+
Vimentin	−	NA	+	NA	+	NA	+	+
Nuclear pleomorphism	+	+	+	+	+	+	+	+
Mitotic index	high	high	high	high	high	high	high	high
Tumor necrosis	moderate	low	low	high	moderate	high	low	high
Invasiveness	moderate	moderate	high	high	moderate	low	low	high
Angiogenesis	high	high	high	moderate	moderate	high	low	high
Immunogenicity	high	high	low	low	low	high	high	high
References	[13,14,15,16,17,18,19,20,21,22,23,24,25]	[13,14,15,16,17,18,19,20,21,22,23,24,25]	[13,14,15,16,17,18,19,20,21,22,23,24,25]	[13,14,15,16,17,18,19,20,21,22,23,24,25]	[15,26,27,28,29,30,31,32,33,34,35,36]	[15,26,27,28,29,30,31,32,33,34,35,36]	[37,38,39,40,41,42,43,44]	[37,38,39,40,41,42,43,44]

“+”, Yes; “−“, No; NA, not available; mut., mutation. Abbreviations: EGFR, epidermal growth factor receptor; GFAP, glial fibrillary acidic protein; IDH, isocitrate dehydrogenase; PDGF, platelet-derived growth factor; PTEN, phosphatase and tensin homolog.

**Table 2 ijms-24-16316-t002:** Summary of in vitro and in vivo models used for glioblastoma research, with the advantages and limitations of each.

Model	Brief Description	Advantages	Limitations	References
In vitro				
Two-dimensional cell culture	The simplest form of growing cell lines in an appropriate medium and passaging them over several generations	- Simple techniques that require basic wet lab skills and minimal costs - Can be used for the preliminary screening of drugs for their effect on the viability, proliferation, and migration of cells using specialized assays	- Only comprised of tumor cells and hence lack a tumor microenvironment and other non-cancer cells - Lack clonal heterogeneity - Prone to genetic drifting and a loss of resemblance to the parent tumor after several generations - Do not reflect the impact of the three-dimensional organization of tumors and the gradient of oxygen and nutrient concentrations	[300,345,346]
Spherical cancer models	Three-dimensional spheroids that originate from glioblastoma cancer stem cells	- Mimic the three-dimensional architecture of tumors with a necrotic center and a peripheral layer of proliferating cells - Retain a degree of spatial organization and tissue polarity - Help in studying tumor–cell interactions in a three-dimensional system - Good for high-throughput screening and personalized therapeutic testing	- Lack non-tumor cells and thus fail to faithfully mirror the tumor microenvironment and the interactions of glioblastoma cells with non-tumor cells - Fail to reproduce intratumoral heterogeneity, especially after a long time in the culture - Cannot reflect the impact of the blood–brain barriers and tumor vascularization on the growth patterns and the response to therapies	[150,334,345,346]
Brain organotypic models	This model aims to act as a miniature of the brain and is derived using pluripotent stem cells. Glioblastoma is introduced via genetic editing or coculturing with patient-derived cancer stem cells.	- Can recapitulate the tumor microenvironment and interactions between tumor and non-tumor cells - Offer a system for characterizing glioblastoma invasion and for developing therapeutics that hinder this process - Mirror the steps of carcinogenesis and enable the study of tumors with customized genetic backgrounds (for genetically engineered models) - Retain the original tumor mutation profile and heterogeneity (for patient-derived models) - Biobanking is possible, allowing for experimental time management	- Poor resemblance to human vascularization processes and the immune environment - Do not faithfully represent the characteristics of the blood–brain–tumor barrier - Significant variability (different success rates depending on the tumor’s IDH profile) - Poor microenvironment retention over long periods - Limited growth extent due to the issues of nutrient and oxygen diffusion to the core	[150,334,345,346]
Scaffolds (3D bioprinting)	This technology refers to the construction of three-dimensional networks of various cells and biomaterials that resemble the constituents of glioblastoma and its microenvironment.	- Can adequately resemble the architecture and makeup of human glioblastoma - Can be used to evaluate the mobility and invasiveness of glioblastoma cells and their interactions with immune and other non-tumor cells - Offer the ability to modulate the stiffness of the extracellular matrix and simulate various conditions	- Expensive and necessitate the availability of a bioprinter - The used bio-inks might interfere with certain physiological processes - Do not contain vasculature - Tumors are printed rather than generated via carcinogenesis and thus do not mirror the steps of glioma-genesis	[150,334,345,346]
Microfluidics	This technology utilizes microchips and micromechanical valves that enable researchers to accurately control the medium constituents and the flow of fluid at the nanoliter level.	- Offer a novel approach for studying the dynamic changes in the tumor microenvironment (hypoxia, metabolite concentrations, etc.) - Allow for the robust screening of pharmacologic treatment candidates - Can accurately recreate human physiologic and pathologic conditions and mimic tissue-specific properties - Can model important processes such as angiogenesis, interstitial fluid flow, and immune interactions - Regulate the placement of cells in different geometries including 3D structures recapitulating the tissue of interest - Parallelization that improves reproducibility - Automation that minimizes human error - Live cell imaging properties - Device design can be tailored to the experimental goals, considering the cells’ shape, size, and density	- Mostly cannot function as stand-alone investigational techniques and need to be integrated with other models - Cannot be used for long-term experiments - Have limitations when it comes to the complexity and size of structures that they can model - Materials used in these systems can absorb hydrophobic molecules and subsequently disrupt the concentrations of some proteins and lipids - PDMS may be toxic to the cells if not properly cured - Prone to channel congestion with cells and valve malfunction	[150,231,290,347]
In vivo				
Allografts in syngeneic models	This model involves the implantation of established cancer cell lines that were generated in a certain species into strain-matched hosts.	- Can be easily generated and maintained compared to other in vivo models - Reproducibility - Well studied and characterized in the literature - Commonly used for immune studies	- Do not recapitulate intratumoral heterogeneity - Differences in the genetic and molecular profile of tumor cells compared to human glioblastoma - Differences in the tumor microenvironment and immune interactions compared to humans	[15,300,334,345]
Patient-derived xenografts in immunocompromised models	This model is created by harvesting glioblastoma cells from cancer patients and implanting them into animal hosts that were engineered to develop deficient immune systems	- Better mirror the characteristics and microenvironment of human glioblastoma - Preserve intratumoral heterogeneity - Allow for the generation of personalized models for patients that can predict the response to treatment and guide clinical management	- Require an immunocompromised host, which limits their ability to recapitulate tumor–immune interactions and prevents their use for immune studies	[15,300,334,345]
Patient-derived xenografts in humanized/immunotolerant models	This model is also generated using glioblastoma cells originating from human patients; however, these cells are implanted into mice that were engrafted with hematopoietic stem cells that can generate a human-like immune system (humanized mice). Alternatively, immunocompetent mice are implanted with tumor cells and treated with immunosuppressive drugs until the tumors are well established (immunotolerant mice).	- Faithful recapitulation of the tumor’s genetic background and intratumoral heterogeneity - Allow for the investigation of interactions between human-derived glioblastoma cells and a human-like immune system (in the humanized model). - Enable testing immunotherapies against human glioblastoma cell lines and patient-derived xenografts.	- Very expensive and require significant expertise - Time-consuming to establish - Follow-up period might be limited by the development of graft versus host disease (in humanized models)	[15,300,334,345]
Genetically engineered and viral vector-induced models	These models rely on the genetic modification of animals in order to turn off/on specific tumor suppressor/oncogenes. The result is an animal model that develops glioblastoma in a “spontaneous” manner.	- Generation does not involve breaching the blood–brain barrier - Better recapitulate the process of carcinogenesis and the development of human glioblastoma - Flexibility to introduce various genetic modifications and investigate their impact - Preserved host immune system	- Have high costs and require significant expertise - Time-consuming to establish - Cannot guarantee uniformity among different animals/experiments	[15,300,334,345]

## Data Availability

Not applicable.

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
