# Peer review of "Preclinical Models and Technologies in Glioblastoma Research: Evolution, Current State, and Future Avenues"

_ijms, 2023, doi:10.3390/ijms242216316_

Round 1

Reviewer 1 Report

Comments and Suggestions for Authors

Comments

The manuscript title “Preclinical models and technologies in glioblastoma research: Evolution, current status quo, and future avenues” is interesting and it under the scope of journal.  The main aim of this is to describe the evolution of in vitro and in vivo preclinical models over the past decades. There are few minor suggestion if authors add it looks more informative and attractive, the details are as under:

1.      Title should replace the Latin phrase “current status quo” with English phrase to more readable attractive.

2.      Authors should add material and methods sections where it should be mentioned about the selection criteria, search, data collection and analysis, and study quality and risk of bias.

3.      Line no. 64 the major heading should be at 2. 0 In- vitro models while the heading (. Cell lines used in GBM research) should be subheading as 2.1.

4.      Similarly heading no 5 (5. Three dimensional in vitro models) line no 636 must be included under heading no 2.0 in-vitro models as a subheading.

5.      Table. 1 must be added extra column for the references.

6.       Heading no 3. Types of animal models used line no262 should be under sub heading In-vivo model.

7.      Table. 2 must be added references in extra column for each observation and for particular models not in foot notes. In this table follow the observation from in-vitro model to in-vivo model.

8.      Write the project number in the heading funding monitory support of this work.

Author Response

We thank the editors and reviewers for taking the time to review our manuscript and for their valuable suggestions and constructive feedback. All changes are highlighted in yellow in the updated manuscript file.

Reviewer 1:

The manuscript title “Preclinical models and technologies in glioblastoma research: Evolution, current status quo, and future avenues” is interesting and it under the scope of journal.  The main aim of this is to describe the evolution of in vitro and in vivo preclinical models over the past decades. There are few minor suggestion if authors add it looks more informative and attractive, the details are as under:

  1. Title should replace the Latin phrase “current status quo” with English phrase to more readable attractive.

Thank you for the suggestion. The Latin phrase “current status quo” was changed to the English one “current state.”

  1. Authors should add material and methods sections where it should be mentioned about the selection criteria, search, data collection and analysis, and study quality and risk of bias.

Thank you for your suggestion. Indeed, adding a methods section can increase the transparency and validity of the manuscript. We added a section “2. Methods” (lines 68 to 79) that outlines our search strategy, screening process, and an assessment of publication bias as requested by the reviewer. Since this manuscript is a narrative review rather than a systematic one, we kept our methods section concise and did not include an extensive assessment of study quality.

  1. Line no. 64 the major heading should be at 2. 0 In- vitro models while the heading (. Cell lines used in GBM research) should be subheading as 2.1.

  1. Similarly heading no 5 (5. Three dimensional in vitro models) line no 636 must be included under heading no 2.0 in-vitro models as a subheading.

  1. Heading no 3. Types of animal models used line no262 should be under sub heading In-vivo model.

For comments 3, 4, and 6: We are not able to put all sections under the two major headings “In vitro models” and “In vivo models,” because if we do this, we will have more subheading levels that permitted by the journal. However, we took the reviewer’s insightful comment into consideration and rearranged the sections of the manuscript as suggested so that the in vitro models (cell lines, three-dimensional models, and microfluidics) are presented first (now sections 3, 4, and 5, respectively), and then the in vivo models (animal types and in vivo model creation) are presented later (now sections 6 and 7).

  1. Table. 1 must be added extra column for the references.

We agree with the reviewer that presenting the references use for Table 1 within the table makes it more credible and useful for readers. So, we added the references into Table 1. However, we added them as a new row rather than a column, because each set of references is specific to a presented cell line.

  1. Table. 2 must be added references in extra column for each observation and for particular models not in foot notes. In this table follow the observation from in-vitro model to in-vivo model.

Thank you for your comment. We followed the reviewer’s suggestion and rearranged the table so that in vitro models are presented first, and in vivo models are presented after, which matches the flow in the manuscript. We also added the references in a new column.

  1. Write the project number in the heading funding monitory support of this work.

The funding for this project is not part of a specific grant. Hence, there is no project number. We just wanted to acknowledge the philanthropic contributions that The Khatib Brain Tumor Center has made to support our laboratory and advance our research in general.

Reviewer 2 Report

Comments and Suggestions for Authors

Authors need to revise the manuscript accordingly.

Author Response

We thank the editors and reviewers for taking the time to review our manuscript and for their valuable suggestions and constructive feedback. All changes are highlighted in yellow in the updated manuscript file.

Reviewer 2:

Overview

The review emphasizes the importance and role of various preclinical models employed in glioblastoma research. The review provides enormous amount of significant information with in depth explanation on cell line and animal models.

Major Comments

  1. Only 49 references out of 331 references cited were published from las three years were used. Include more recent articles.

Thank you for pointing this out. We agree that more recent references need to be included to ensure that the manuscript is up-to-date and that it offers a comprehensive and representative review of the literature. Hence, we ran our search strategy again on November 3rd, 2023, and made sure to update our references and include the latest findings on the topic. We successfully increased the number of references published after 2020 to 68 references (18% of all references). Moreover, the percentage of papers cited that were published in the last six years (2017 and beyond) increased to 33%. The added references are highlighted in yellow in the reference list, as well as their corresponding citations in the text.

The fact that this review offers an overview of the historic evolution of glioblastoma models makes it difficult to obtain higher percentages of recent publications. So, please take this into consideration.

  1. Tabulate the various animal models used. Their advantages, disadvantages, possible short comings.

Thank you for the valuable suggestion. We agree that a summary of the advantages and disadvantages of the animal models is important for the review. So, we presented this information in Figure 2 (page 20) along with the illustrations of the different models. We believe that this might be more appealing for the readers.

  1. Include more figures.

We added two more figures. Figure 1 (page 8) now illustrates the different in vitro models used (two-dimensional, three-dimensional, and microfluidics). Figure 2 (page 20) illustrates the different animals used for in vivo research and discusses the advantages and shortcomings of each.

  1. Include statistical data as to prevalence, incidence, and mortality of glioblastoma across the world.

Thank you for your suggestion. We added more details regarding the overall incidence, age-specific incidence, and gender differences for glioblastoma. The new sentences were added in the introduction (lines 35 to 37).

Minor Comments

  1. Too many abbreviations. Provide the list in the beginning.

We added a list of abbreviations as suggested by the reviewer. We added the list at the end of the manuscript rather than at the beginning in order not to disrupt the title page formatting required by the journal.

Remark

The review informative and requires only few changes. Recently publications should be included (possibly on or after 2020).

Reviewer 3 Report

Comments and Suggestions for Authors

Dear author,

My comment about your manuscript is that it should be modified so that it is easier to read by the general public.

1. Please, highlight the novelty of your manuscript in the abstract and introduction section with the latest reference.

2. Please add in the short advantages and disadvantages of cell lines in GBM research with the latest reference. If it is possible, make a schematic of this section.

3. Please change the title "Types of animal models used" and make a schematic of this section. 

4. Please make a schematic of sections 5, and 6 and make it short and consistent.

5. Please add some more recent references.

Best Regards

Author Response

We thank the editors and reviewers for taking the time to review our manuscript and for their valuable suggestions and constructive feedback. All changes are highlighted in yellow in the updated manuscript file.

Reviewer 3:

My comment about your manuscript is that it should be modified so that it is easier to read by the general public.

  1. Please, highlight the novelty of your manuscript in the abstract and introduction section with the latest reference.

Thank you for your suggestion. We have included a sentence in each of the abstract (line 26) and introduction (line 68 to 71) to further highlight the novelty of the manuscript and the added-value that it offers for the field of glioblastoma research.

  1. Please add in the short advantages and disadvantages of cell lines in GBM research with the latest reference. If it is possible, make a schematic of this section.

Thank you for your suggestion. We added the advantages and disadvantages of cell lines in GBM research in Table 2 along with the other models.

  1. Please change the title "Types of animal models used" and make a schematic of this section.

The title was changes to “Animal used for in vivo GBM research.” We have also created a schematic of this section as requested. The new Figure 2 (page 20) illustrates the different animal models and discusses the advantages and disadvantages of each.

  1. Please make a schematic of sections 5, and 6 and make it short and consistent.

Thank you for the suggestion. As per your recommendation, we have added a schematic that represents all in vitro models used for glioblastoma research. Please see the new Figure 1 (page 8).

  1. Please add some more recent references.

We agree with the reviewer that the manuscript can become more comprehensive and representative if more updated references were added. Hence, we performed another search of the literature and made sure to update our references and include any new findings that are relevant to our manuscript. We were able to increase the number of references published within the last three years (2020 and beyond) to 68 and of those published within the last six years (2017 and beyond) to 122. The added references are highlighted in yellow in the references list, along with their corresponding in-text citation in the main body.